# Contrast and Mix: Temporal Contrastive Video Domain Adaptation with Background Mixing

**Aadarsh Sahoo**[1]    **Rutav Shah**[1]    **Rameswar Panda**[2]    **Kate Saenko**[2,3]    **Abir Das**[1]

[1] IIT Kharagpur, [2] MIT-IBM Watson AI Lab, [3] Boston University

{sahoo_aadarsh@, rutavms@, abir@cse.}iitkgp.ac.in, rpanda@ibm.com, saenko@bu.edu

## Abstract

Unsupervised domain adaptation which aims to adapt models trained on a labeled source domain to a completely unlabeled target domain has attracted much attention in recent years. While many domain adaptation techniques have been proposed for images, the problem of unsupervised domain adaptation in videos remains largely underexplored. In this paper, we introduce Contrast and Mix (CoMix), a new contrastive learning framework that aims to learn discriminative invariant feature representations for unsupervised video domain adaptation. First, unlike existing methods that rely on adversarial learning for feature alignment, we utilize temporal contrastive learning to bridge the domain gap by maximizing the similarity between encoded representations of an unlabeled video at two different speeds as well as minimizing the similarity between different videos played at different speeds. Second, we propose a novel extension to the temporal contrastive loss by using background mixing that allows additional positives per anchor, thus adapting contrastive learning to leverage action semantics shared across both domains. Moreover, we also integrate a supervised contrastive learning objective using target pseudo-labels to enhance discriminability of the latent space for video domain adaptation. Extensive experiments on several benchmark datasets demonstrate the superiority of our proposed approach over state-of-the-art methods. Project page: https://cvir.github.io/projects/comix.

## 1   Introduction

Unsupervised domain adaptation (UDA), which alleviates the requirement of large amounts of annotated data by adapting a model learned on a labelled source domain to an unlabelled target domain, has drawn a great deal of attention in the last few years [12, 80]. Much progress has been made in developing deep UDA methods by minimizing the cross-domain divergence [39, 70], adding adversarial domain discriminators [20, 74], and image-to-image translation techniques [26, 51]. However, despite impressive results on commonly used benchmark datasets (*e.g.*, [61, 75, 57]), most of the methods have been developed only for images and not for videos, where the annotation task is often more complicated requiring tedious human labor in comparison to images.

More recently, very few works have attempted deep UDA for video action recognition by directly matching segment-level features [8, 27, 50, 42] or with attention weights [11, 53]. However, (1) trivially matching segment-level feature distributions by extending the image-specific approaches, without considering the rich temporal information may not alone be sufficient for video domain adaptation; (2) prior methods often focus on aligning target features with source, rather than exploiting any action semantics shared across both domains (*e.g.*, difference in background with the same action: videos in the top row of Figure 1 are from the source and target domain respectively, but both capture the same action *walking*); (3) existing methods often rely on complex adversarial learning which is unwieldy to train, resulting in very fragile convergence.

35th Conference on Neural Information Processing Systems (NeurIPS 2021).

Meanwhile, self-supervised pretext tasks like predicting rotation and translation have recently emerged as an alternative to adversarial learning for unsupervised domain adaptation in images [38, 71]. While these works show the promising potential of self-supervised learning in aligning source and target domains, the more recent very successful contrastive representation learning [9, 23, 52] has never been used to adapt video action recognition models to target domains. Motivated by this, in this paper, we explore the following natural, yet important question: *whether and how contrastive learning could be exploited for the challenging and practically important task of unsupervised video domain adaptation for human action recognition?*

To this end, we introduce Contrast and Mix (`CoMix`), a simple yet effective approach based on contrastive learning to adapt video action recognition models trained on a labeled source domain to unlabelled target domains. First, we propose to represent video as a graph and then utilize temporal contrastive self-supervised learning over the graph representations as a nexus between source and target domains to align features, without requiring any additional adversarial learning, as most prior works do in video domain adaptation [8, 11, 53]. Specifically, we maximize the similarity between encoded representations of the same video at two different speeds as well as minimize the similarity between different videos played at different speeds, leveraging the fact that changing video speed does not change an action on both domains. While minimization of contrastive self-supervised losses in both domains simultaneously helps in domain alignment, it ignores

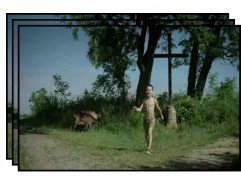 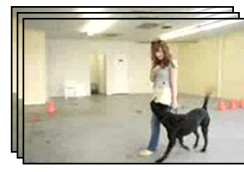

**Source**          **Target**

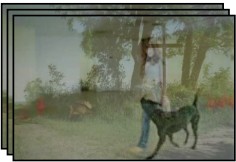

**Mix (Source, Target BG)**    **Mix (Target, Source BG)**

Figure 1: **Background Mixing.** Top row shows two representative videos from the source and target domain respectively. Both videos capture the same action "*walking*" with different backgrounds. Bottom row shows videos obtained after mixing target background with source video and vice versa.

action semantics shared across them as the loss treats each domain individually. To alleviate this, we incorporate new synthetic videos into the temporal contrastive objective, which are obtained by mixing background of a video from one domain to a video from another domain, as shown in Figure 1 (bottom). Importantly, since mixing background doesn't change the temporal dynamics, we introduce pseudo-labels for the mixed videos to be same as the label of the original videos and consider additional positives per anchor (see Figure 2), which encourages the model to generalize to new samples that may not be covered by temporal contrastive learning in hand. In other words, mixed background video of an input sample in the embedding space act as small semantic perturbations that are not imaginary, i.e., they are representative of the action semantics shared across source and target domains. Finally, rather than relying only on the supervision of source categories to learn a discriminative representation, we generate pseudo-labels for the target samples in every batch and then harness the label information using a temporal supervised contrastive term, that pushes the examples from the same class close and the examples from different classes further apart (Figure 2: right). While our modified contrastive losses are motivated by the supervised contrastive learning [30], we use pseudo labels for exploiting shared action semantics and discriminative information from target domain, instead of using true labels as an alternative to supervised cross-entropy loss (which is not present for target samples). To the best of our knowledge, ours is the first work that successfully leverages contrastive learning in an unified framework to align cross-domain features while enhancing discriminabilty of the latent space for unsupervised video domain adaptation.

To summarize, the main **contributions** of our work are as follows:

- We introduce Contrast and Mix (`CoMix`), a new contrastive learning framework to learn discriminative invariant feature representations for unsupervised video domain adaptation. Overall, `CoMix` is simple and easy to implement which perfectly fits into modern mini-batch end-to-end training.

- We propose a novel extension to temporal contrastive loss by using background mixing that allows additional positives per anchor, thus adapting contrastive learning to leverage action semantics shared across both domains. We also integrate a supervised contrastive learning objective using pseudo label information from the target domain to enhance discriminabilty of the latent space.

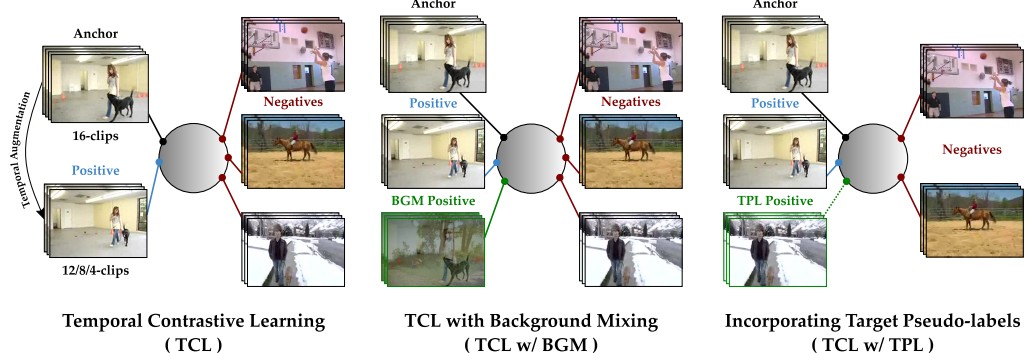

**Temporal Contrastive Learning
( TCL )**

**TCL with Background Mixing
( TCL w/ BGM )**

**Incorporating Target Pseudo-labels
( TCL w/ TPL )**

Figure 2: **Temporal Contrastive Learning with Background Mixing and Target Pseudo-labels.** Temporal contrastive loss (left) contrasts a single temporally augmented positive (same video, different speed) per anchor against rest of the videos in a mini-batch as negatives. Incorporating background mixing (middle) provides additional positives per anchor possessing same action semantics with a different background alleviating background shift across domains. Incorporating target pseudo-labels (right) additionally enhances the discriminabilty by contrasting the target videos with the same pseudo-label as positives against rest of the videos as negatives.

- We conduct extensive experiments on several challenging benchmarks (UCF-HMDB [8], Jester [53], and Epic-Kitchens [50]) for video domain adaptation to demonstrate the superiority of our approach over state-of-the-art methods. Our experiments show that `CoMix` delivers a significant performance increase over the compared methods, *e.g.*, `CoMix` outperforms SAVA [11] (ECCV'20) by $3.6\%$ on UCF-HMDB [8] and TA$^3$N [8] (ICCV'19) by $9.2\%$ on Jester [45] benchmark respectively).

## 2    Related Work

**Action Recognition.** Much progress has been made in developing a variety of ways to recognize video actions, by either applying 2D-CNNs [6, 37, 47, 79] or 3D-CNNs [4, 17, 22, 73]. Many successful architectures are usually based on the two-stream model [67], processing RGB frames and optical-flow in two separate CNNs with a late fusion in the upper layers [29]. SlowFast network [18] employs two pathways for recognizing actions by processing a video at different frame rates. Mitigating background bias in action recognition has also been presented in [10, 36]. Despite remarkable progress, these models critically depend on large labeled datasets which impose challenges for cross-domain action recognition. In contrast, our work focuses on unsupervised domain adaptation for action recognition, with labeled data in source domain, but only unlabeled data in target domain.

**Unsupervised Domain Adaptation.** Unsupervised domain adaptation has been studied from multiple perspectives (see reviews [12, 80]). Representative works minimize some measurement of distributional discrepancy [21, 39, 65, 70] or adopt adversarial learning [5, 20, 40, 56, 74] to generate domain-invariant features. Leveraging image translation [25, 26, 51] or style transfer [15, 91] is also another popular trend in domain adaptation. Deep self-training that focus on iteratively training the model using both labeled source data and generated target pseudo labels have been proposed in [46, 90]. Semi-supervised domain adaptation leveraging a few labeled samples from the target domain has also been proposed for many applications [14, 33, 63]. A very few methods have recently attempted video domain adaptation, using adversarial learning combined with temporal attention [8, 42, 53], multi-modal cues [50], and clip order prediction [11]. While existing video DA methods mainly rely on adversarial learning (which is often complicated and hard to train) in some form or other, they do not take any action semantics shared across domains into consideration. Our approach on the other hand, successfully leverages temporal contrastive learning to learn domain-invariant features while exploiting shared action semantics through background mixing for video domain adaptation. Recently, self-supervised tasks like predicting rotation and translation have been used for unsupervised domain adaptation and generalization, mainly for images [3, 38, 71]. By contrast, we focus on the more challenging problem of domain adaptation for human action recognition, where our goal is to align domains by learning consistent features representing different speeds of unlabeled videos. We further propose a temporal supervised contrastive loss to ensure discriminabilty by considering pseudo-labeling in an unified framework for video domain adaptation.

**Contrastive Learning.** Contrastive representation learning is becoming increasingly attractive due to its great potential to leverage large amount of unlabeled images [9, 16, 23, 48, 24, 52] and videos [19, 31, 54, 60, 59, 78]. Speed of a video is investigated for self-supervised [1, 28, 77, 86] and

semi-supervised learning [68, 94] unlike the problem we consider in this paper. Recent works [84, 87] utilize contrastive learning with different augmentations for learning unsupervised representations of graph data. Contrastive learning has also been recently used in supervised settings, where labels are used to guide the choice of positive and negative pairs [30]. While our approach is inspired by these, we propose a novel temporal contrastive learning framework with background mixing for video domain adaptation, which to our best knowledge has not been explored in the literature.

**Image Mixtures.** Mixup regularization [89] and its variants [2, 76, 88] that train models on virtual examples constructed as convex combinations of pairs of images and labels have been used to improve the generalization of neural networks. Very few methods apply Mixup in domain adaptation, but mainly to stabilize the domain discriminator [62, 83, 85] or to smoothen the predictions [44]. Several works have recently leveraged the idea of different image mixtures [34, 66] for improving contrastive representation learning. Our proposed background mixing can be regarded as an extension of this line of research by adding background of a video from one domain to a video from another domain, to explore shared semantics while learning domain-invariant features for action recognition.

## 3 Proposed Method

Unsupervised video domain adaptation aims to improve the model generalization performance by transferring knowledge from a labeled source domain to an unlabeled target domain. Formally, we have a set of labelled source videos $\mathcal{D}_{source} = \{(\mathbf{V}^{i\{s\}}, y^i)\}_{i=1}^{N_S}$ and a set of unlabelled target videos $\mathcal{D}_{target} = \{\mathbf{V}^{i\{t\}}\}_{i=1}^{N_T}$, with a common label space $\mathcal{L}$. Given these data sets, our goal is to learn a single model for action recognition that performs well on previously unseen target domain videos.

**Approach Overview.** Figure 3 illustrates an overview of CoMix. Our action recognition model consists of a feature encoder $\mathcal{F}$ with a temporal graph encoder $\mathcal{G}$. Given a video, the feature encoder $\mathcal{F}$ first extracts clip-level features, and then a graph encoder $\mathcal{G}$ utilizes those features to model intrinsic temporal relations for providing a robust encoded representation for action recognition. CoMix adopts supervised learning on the source videos, as the labels are available, jointly with two novel temporal contrastive learning loss terms to align the features for domain adaptation. Specifically, we maximize the similarity of the encoded representation of the fast version of a video (represented by $f$ clips) with that of the slow version of the same video (represented by $s$ clips, where $s < f$) as well as minimize the similarity of the representations of different videos within each of the two domains. However, as temporal contrastive loss treats each domain individually, we further add

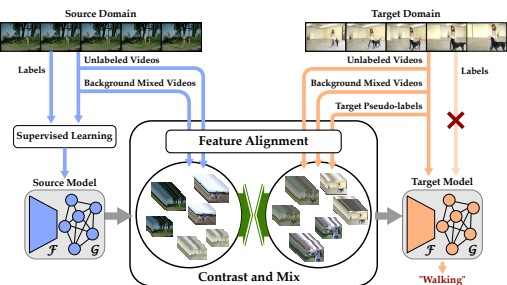

Figure 3: **An Overview of our Approach.** Given labeled videos in source domain and only unlabeled videos in target domain, CoMix adopts supervised learning on source videos, jointly with temporal contrastive learning on both domains to align features. Additional cross-domain contrastive supervision is obtained using background mixing across domains and using target pseudo-labels for enhancing discriminability of the latent space. CoMix provides a more simpler yet effective approach than adversarial learning for aligning both domains.

two new sets of synthetic videos that contain source videos mixed with target background and vice versa, respectively for introducing the background variations among the videos while keeping the action semantics intact. Finally, we generate pseudo-labels for the target videos in every mini-batch and utilize them using another temporal supervised contrastive term. This term contrasts target videos with the same pseudo-label as positives to learn features discriminative for the target domain. We now describe each of our proposed components individually in detail in the following subsections.

**Video Representation.** Capturing long-range temporal structure in videos is crucial for action recognition, which in turn affects the overall generalization performance of a model when adapting across domains. Thus, we adopt a graph convolutional neural network ($\mathcal{G}$) on top of a 3D convolutional neural network ($\mathcal{F}$) as our video feature encoder. Specifically, for a video $\mathbf{V}$ with $n$ clips, the feature extractor $\mathcal{F}$ maps the clips into the corresponding sequence of features, which alone do not incorporate the rich temporal structure of the video. Therefore, we use the temporal graph encoder which constructs a fully connected graph on top of the clip-level features, with learnable edge weights through a parameterized adjacency matrix, as in [81]. With these graph representations, we apply a graph convolutional neural network with three layers and finally perform average pooling over all

the node features to output the encoded representation of the video $\mathbf{V}$. In summary, the end-to-end network $\mathcal{G}(\mathcal{F}(.)) : \mathbf{V} \to \mathbb{R}^c$ takes a sequence of clips from a video as input and outputs confidence scores (logits) over the number of classes $c$ for recognizing actions.

**Temporal Contrastive Learning.** Given video representations, our goal is to leverage contrastive self-supervised learning in both domains for unsupervised domain adaptation. To this end, we use temporal speed invariance in videos as a proxy task and enforce this with a pairwise contrastive loss. Specifically, our key idea is to represent videos in two different temporal speeds (fast and slow) to obtain their encoded representations and then consider the fast and slow version representations of the same video to constitute positive pairs, while versions from different videos constitute negative pairs. Formally, let us consider a mini-batch of $B$ videos $\{\mathbf{V}_n^1, \mathbf{V}_n^2, ..., \mathbf{V}_n^B\}$ with corresponding feature representations $\{\mathbf{z}_n^1, \mathbf{z}_n^2, ..., \mathbf{z}_n^B\}$, where each of the videos $\mathbf{V}_n^i$ is represented using $n$ number of sampled clips. Let $f$ be the number of clips used to represent the fast version of the videos (forwarded through the base branch), and $s$ be that used for the slow version (forwarded through the auxiliary branch), with $s < f$, as shown in Figure 4. Given positive and negative pairs, the model is trained such that it learns to maximize agreement between positive pairs, while minimizing agreement between negative pairs. This is achieved by employing a temporal contrastive loss ($\mathcal{L}_{tcl}$) as

$$\mathcal{L}_{tcl}(\mathbf{V}_f^i, \mathbf{V}_s^i) = -\log \frac{h(\mathbf{z}_f^i, \mathbf{z}_s^i)}{h(\mathbf{z}_f^i, \mathbf{z}_s^i) + \sum_{\substack{j=1, j \neq i \\ v \in \{s,f\}}}^{B} h(\mathbf{z}_f^i, \mathbf{z}_v^j)} \quad (1)$$

where, $h(\mathbf{u}, \mathbf{v}) = \exp(\frac{\mathbf{u}^\top \mathbf{v}}{\|\mathbf{u}\|_2 \|\mathbf{v}\|_2} / \tau)$ is the exponential of cosine similarity measure and $\tau$ is the temperature hyperparameter [9]. We use $f = 16$, and choose $s$ from $\{12, 8, 4\}$ following a random uniform distribution in every training iteration where randomness encourages the model to learn from a variety of temporal speed variations to learn robust representations.

**Background Mixing.** As temporal contrastive loss treats each domain individually, it ignores shared action semantics which is vital for domain alignment. Thus, we propose a new perspective of temporal contrastive loss through background mixing, specifically to alleviate the cross-domain background shift, as seen in Figure 1. The basic idea is to obtain the background frames for the videos in one domain and mix it with the frames of the videos from the other domain. More details on how we extract the backgrounds are provided in supplementary material. This introduces variation in each of the domains by adding new synthetic videos with the same action semantics as earlier, but possessing background from the other domain. Given two videos $\mathbf{V}^{i\{s\}} \in \mathcal{D}_{source}$ and $\mathbf{V}^{i\{t\}} \in \mathcal{D}_{target}$ with corresponding background frames (single image per video) as $\mathbf{BG}^{i\{s\}}$ and $\mathbf{BG}^{i\{t\}}$, we obtain the synthetic videos in both domains by a convex combination of the background with each of the frames in the videos as follows.

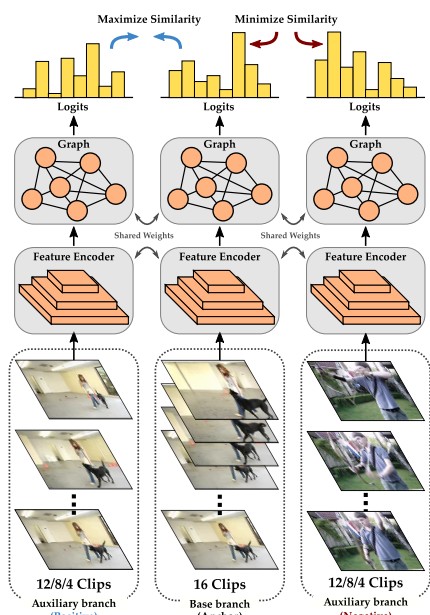

Figure 4: **Temporal Contrastive Loss.** Given unlabeled videos, we maximize similarity between encoded representations of the same video at two different speeds (fast and slow) as well as minimize similarity between different videos played at different speeds.

$$\hat{\mathbf{V}}^{i\{s\}} = (1 - \lambda) \cdot \mathbf{V}^{i\{s\}} + \lambda \cdot \mathbf{BG}^{i\{t\}}$$
$$\hat{\mathbf{V}}^{i\{t\}} = (1 - \lambda) \cdot \mathbf{V}^{i\{t\}} + \lambda \cdot \mathbf{BG}^{i\{s\}} \quad (2)$$

where, $\lambda$ is sampled from the uniform distribution $[0, \gamma]$, $\hat{\mathbf{V}}^{i\{s\}}$ and $\hat{\mathbf{V}}^{i\{t\}}$ correspond to the video from source domain with target background and vice versa, respectively. The main operation in our proposed background mixing is to generate a synthetic video with background from the other domain while retaining the temporal action semantics intact. Since mixing background doesn't change the motion pattern of a video which actually defines an action, we assume both the original and mixed video to be of the same action class and go beyond single instance positives in Eq. 1 by adding additional positives per anchor, as in supervised contrastive learning [30] (see Figure 2 for an illustrative example). The modified temporal contrastive

loss with background mixing ($\mathcal{L}_{bgm}$) is defined as below:

$$\mathcal{L}_{bgm}(\mathbf{V}_f^i, \mathbf{V}_s^i) = -\frac{1}{|\mathbf{P}(\mathbf{z}_f^i)|} \sum_{\mathbf{p} \in \mathbf{P}(\mathbf{z}_f^i)} \log \frac{h(\mathbf{z}_f^i, \mathbf{p})}{\sum\limits_{\mathbf{p} \in \mathbf{P}(\mathbf{z}_f^i)} h(\mathbf{z}_f^i, \mathbf{p}) + \sum\limits_{\substack{j=1, j \neq i \\ v \in \{s,f\}}}^{B} \{h(\mathbf{z}_f^i, \mathbf{z}_v^j) + h(\mathbf{z}_f^i, \hat{\mathbf{z}}_v^j)\}}$$

(3)

where, $\mathbf{P}(\mathbf{z}_f^i) \equiv \{\mathbf{z}_s^i, \hat{\mathbf{z}}_s^i, \hat{\mathbf{z}}_f^i\}$ is the set of positives for the anchor $\mathbf{z}_f^i$, and $\hat{\mathbf{z}}_{s/f}^i$ represent the feature representation of the corresponding background-mixed video depending on the domain to which $\mathbf{V}^i$ belongs. Note that for anchor $\mathbf{z}_f^i$, there are 3 positive pairs: (a) slow version of the mixed video ($\hat{\mathbf{z}}_s^i$), (b) fast version of the mixed video ($\hat{\mathbf{z}}_f^i$), and (c) slow version of the original video ($\mathbf{z}_s^i$). Also, the loss is computed for all positive pairs in the mini-batch, i.e., $(\mathbf{V}_f^i, \mathbf{V}_s^i)$, $(\mathbf{V}_s^i, \mathbf{V}_f^i)$, $(\hat{\mathbf{V}}_f^i, \hat{\mathbf{V}}_s^i)$, and $(\hat{\mathbf{V}}_s^i, \hat{\mathbf{V}}_f^i)$. Simultaneous minimization of $\mathcal{L}_{bgm}$ in both source and target domains not only learns temporal dynamics but also helps to better align the features for video domain adaptation by leveraging action semantics shared across both domains. Our background mixing is especially effective in video domain adaptation as it enforces the model to be robust to domain changes (i.e., difference in background as shown in Figure 1) while leaving the action semantics intact. Further, it can also be adopted as a data augmentation strategy for improved generalization in standard video action recognition: we leave this as an interesting future work.

**Incorporating Target Pseudo Labels.** While temporal contrastive loss with background mixing helps in aligning the learned representations across the two domains, we cannot fully rely on source categories to learn features discriminative for target domain. Therefore, we propose to use a supervised contrastive loss [30] over pseudo-labeled target samples, an extended version of temporal contrastive loss in Eqn. 1 to enhance discriminabilty by allowing many samples per anchor to be positive, so that videos of the same pseudo-label can be attracted to each other in the embedding space. Let $A$ be the subset of videos assigned pseudo-labels using a confidence threshold, from a mini-batch of $B$ videos, the supervised temporal contrastive loss for incorporating target pseudo-labels ($\mathcal{L}_{tpl}$) is defined as

$$\mathcal{L}_{tpl}(\mathbf{V}_f^i, \mathbf{V}_s^i) = -\frac{1}{|\mathbf{P}(\mathbf{z}_f^i)|} \sum_{\mathbf{p} \in \mathbf{P}(\mathbf{z}_f^i)} \log \frac{h(\mathbf{z}_f^i, \mathbf{p})}{\sum\limits_{\mathbf{p} \in \mathbf{P}(\mathbf{z}_f^i)} h(\mathbf{z}_f^i, \mathbf{p}) + \sum\limits_{\substack{a \in A, a \neq i \\ v \in \{s,f\}}} h(\mathbf{z}_f^i, \mathbf{z}_v^a)}$$

(4)

where, $\mathbf{P}(\mathbf{z}_f^i) \equiv \{\mathbf{z}_s^p, \mathbf{z}_f^p : p \in A \, \& \, \tilde{y}^p = \tilde{y}^i\} \setminus \{\mathbf{z}_f^i\}$ is the set of all positives for video $\mathbf{V}_f^i$ and $\tilde{y}^i$ represent the pseudo-label for target video $\mathbf{V}^i$. Note that the set of positives ($\mathbf{P}(.)$) includes all the target domain samples (fast and slow) classified as the same action class as that of the anchor ($\mathbf{z}_f^i$) through the pseudo labels. Following [95], we leverage a temporal ensemble prediction for a given video $\mathbf{V}^i$ from the target domain to produce robust and better-calibrated version of pseudo-labels. Specifically, we obtain the encoded (logits) representations $\mathbf{z}_f^i$ and $\mathbf{z}_s^i$ from the base and auxiliary branch respectively and then compute the pseudo-label as $\tilde{y}^i = \arg\max_k \text{softmax}(\mathbf{z}_{fused}^i)$, where $\mathbf{z}_{fused}^i$ represents the mean of both logits. We consider the class index $k$ on which the model is most confident among $c$ classes, provided it is higher than a confidence threshold.

**Optimization.** Besides the losses $\mathcal{L}_{bgm}$ and $\mathcal{L}_{tpl}$, we minimize the standard supervised cross-entropy loss ($\mathcal{L}_{ce}$) on the labelled source videos as follows.

$$\mathcal{L}_{ce}(\mathbf{V}^{i\{s\}}, y^i) = -\sum_{k=1}^{c} (y^i)_k \log(\mathcal{G}(\mathcal{F}(\mathbf{V}^{i\{s\}})))_k$$

(5)

Overall, the loss function for training our model involving both source and target domain data is,

$$\mathcal{L}_{CoMix} = \mathcal{L}_{ce}^{\{s\}} + \lambda_{bgm}(\mathcal{L}_{bgm}^{\{s\}} + \mathcal{L}_{bgm}^{\{t\}}) + \lambda_{tpl}\mathcal{L}_{tpl}^{\{t\}}$$

(6)

where $\lambda_{bgm}$ and $\lambda_{tpl}$ are weights to balance the impact of individual loss terms. To reduce the number of hyper-parameters, we use the same weight $\lambda_{bgm}$ for both $\mathcal{L}_{bgm}^{\{s\}}$ and $\mathcal{L}_{bgm}^{\{t\}}$. Notably, for the semi-supervised domain adaptation setting, we also use supervised cross-entropy loss for the few labeled target domain videos in addition to the source domain videos.

# 4 Experiments

**Datasets.** We evaluate the performance of our approach using several publicly available benchmark datasets for video domain adaptation, namely UCF-HMDB [7], Jester [53], and Epic-Kitchens [50]. UCF-HMDB (assembled by authors in [7]) is an overlapped subset of the original UCF [69] and HMDB datasets [32], containing 3, 209 videos across 12 classes. Jester (assembled by authors in [53]) is a large-scale cross-domain dataset that contains videos of humans performing hand gestures [45] from two domains, namely Source and Target that contain 51, 498 and 51, 415 video clips respectively across 7 classes. Epic-Kitchens (assembled by authors in [50]) is a challenging egocentric dataset that consists of videos across 8 largest action classes from three domains, namely D1, D2 and D3, corresponding to P08, P01 and P22 kitchens on the full Epic-Kitchens dataset [13]. We use the standard training and testing splits provided by the authors in [7, 53, 50] to conduct our experiments on each dataset. More details about the datasets can be found in the supplementary material.

**Baselines.** We compare our approach with the following baselines. (1) source only (a lower bound) and supervised target only (an upper bound) baselines that trains the network using labeled source data and labeled target data respectively, (2) popular UDA methods based on adversarial learning (e.g., DANN [20], and ADDA [74]), (3) existing video domain adaptation methods, including SAVA [11], TA$^3$N [8], ABG [42] and TCoN [53]. We also compare with Source + Target (which simply uses all labelled data available to it to train the network) and ENT [63] in semi-supervised domain adaptation experiments. We directly quote the numbers reported in published papers when possible and use source code made publicly available by the authors of TA$^3$N [8] on both Jester and Epic-Kitchens.

**Implementation Details.** Following [11], we use I3D [4] as the backbone feature encoder network, initialized with Kinetics pre-trained weights. For the temporal graph encoder, we use a 3-layer GCN similar to [81]. We follow the standard 'pre-train then adapt' procedure used in prior works [74, 11] and train the model with only source data to provide a warmstart before the proposed approach is employed. The dimension of the features extracted from the I3D encoder is 1024 which is the same as the node-feature dimension of the initial layer of the GCN. The final layer of the GCN has its node-feature dimension same as the number of action classes in a dataset and uses a mean aggregation strategy to output the logits. We use a clip-length of 8-frames and train all the models end-to-end using SGD with a momentum of 0.9 and a weight decay of 1e-7. We use an initial learning rate of 0.001 for the I3D and 0.01 for the GCN in all our experiments. We use a batch size of 40 equally split over the two domains, where each batch consists of $n$ clips from the same video, where $n$ is 16 for the fast version ($f$) and 12, 8, or 4 for the slow version ($s$). For inference, we use 16 uniformly sampled clips per video and use the base branch of the model to recognize the action. The temperature parameter is set to $\tau = 0.5$. We extract backgrounds from videos using temporal median filtering [58] and empirically set $\gamma = 0.5$ for background mixing. We use a pseudo-label threshold of 0.7 in all our experiments and smooth the cross-entropy loss with $\epsilon = 0.1$, following [72, 49]. We set $\lambda_{bgm}$ and $\lambda_{tpl}$ from $\{0.01, 0.1\}$ depending on the dataset. We report the average action recognition accuracy over 3 random trials. We use 6 NVIDIA Tesla V100 GPUs for training all our models.

**Results on UCF-HMDB.** Table 1 shows results of our method and other competing approaches on UCF-HMDB dataset. Our CoMix framework achieves the best average performance of **90.3%**, which is about **2.2%** more than the previous state-of-the-art performance on this dataset. While comparing with the recent method, SAVA [11] using the same I3D backbone, CoMix obtains **4.5%** and **2.7%** improvement on UCF→HMDB and HMDB→UCF task respectively, without relying on frame attention or adversarial learning.

Table 1: **Results on UCF-HMDB Dataset.** CoMix establishes new state-of-the-art for unsupervised video domain adaptation on UCF-HMDB, by significantly outperforming existing methods.

| Method | Backbone | UCF→HMDB | HMDB→UCF | Average |
|---|---|---|---|---|
| DANN [20] | ResNet-101 | 75.3 | 76.4 | 75.8 |
| JAN [41] | ResNet-101 | 74.7 | 79.3 | 77.0 |
| AdaBN [35] | ResNet-101 | 75.5 | 77.4 | 76.4 |
| MCD [64] | ResNet-101 | 74.4 | 79.3 | 76.8 |
| TA$^3$N [8] | ResNet-101 | 78.3 | 81.8 | 80.1 |
| ABG [42] | ResNet-101 | 79.1 | 85.1 | 82.1 |
| TCoN [53] | ResNet-101 | 87.2 | 89.1 | 88.1 |
| Source Only | I3D | 80.3 | 88.8 | 84.5 |
| DANN [20] | I3D | 80.7 | 88.0 | 84.3 |
| ADDA [74] | I3D | 79.1 | 88.4 | 83.7 |
| TA$^3$N [8] | I3D | 81.4 | 90.5 | 85.9 |
| SAVA [11] | I3D | 82.2 | 91.2 | 86.7 |
| CoMix | I3D | **86.7** | **93.9** | **90.3** |
| Supervised Target | I3D | 95.0 | 96.8 | 95.9 |

These improvements clearly show that our temporal graph contrastive learning with background mixing is not only able to better leverage the temporal information but also shared action semantics, essential for effective video domain adaptation. In summary, CoMix outperforms all the existing video

Table 2: **Results on Jester and Epic-Kitchens Datasets.** CoMix outperforms TA³N [8] by 9.2% on the challenging Jester dataset. On Epic-Kitchens, CoMix achieves the best performance on 5 out of 6 transfer tasks including the best average performance among all compared methods.

| Method | Backbone | Jester | Epic-Kitchens | | | | | | Average |
|--------|----------|--------|------|------|------|------|------|------|---------|
| | | Source→Target | D2→D1 | D3→D1 | D1→D2 | D3→D2 | D1→D3 | D2→D3 | |
| Source Only | I3D | 51.5 | 35.4 | 34.6 | 32.8 | 35.8 | 34.1 | 39.1 | 35.3 |
| DANN [20] | I3D | 55.4 | 38.3 | 38.8 | 37.7 | 42.1 | 36.6 | 41.9 | 39.2 |
| ADDA [74] | I3D | 52.3 | 36.3 | 36.1 | 35.4 | 41.4 | 34.9 | 40.8 | 37.4 |
| TA³N [8] | I3D | 55.5 | **40.9** | 39.9 | 34.2 | 44.2 | 37.4 | 42.8 | 39.9 |
| CoMix | I3D | **64.7** | 38.6 | **42.3** | **42.9** | **49.2** | **40.9** | **45.2** | **43.2** |
| Supervised Target | I3D | 95.6 | 57.0 | 57.0 | 64.0 | 64.0 | 63.7 | 63.7 | 61.5 |

DA methods on UCF-HMDB, showing the efficacy of our approach in learning more transferable features for cross-domain action recognition without using any target labels.

**Results on Jester and Epic-Kitchens.** On the large-scale Jester dataset, our proposed approach, CoMix also outperforms other DA approaches by increasing the Source Only (no adaptation) accuracy from **51.5%** to **64.7%**, as shown in Table 2 (left). In particular, our approach achieves an absolute improvement of **9.2%** over TA³N [8], which corroborates the fact that CoMix can well handle not only the appearance gap but also the action gap present on this dataset (e.g., for the action class "rolling hand", source domain contains videos of "rolling hand forward", while the target domain only consists of videos of "rolling hand backward"). Table 2 (right) summarizes the results on Epic-Kitchens, which is another challenging dataset consisting of total 6 transfer tasks with a large imbalance across different action classes. Overall, CoMix obtains the best on 5 tasks including the best average performance of **43.2%**, compared to only **35.3%** and **39.9%** achieved by the source only and TA³N [8] respectively. While the improvements achieved by our approach are encouraging on both Jester and Epic-Kitchens, the accuracy gap between CoMix and supervised target is still significant (**30.9%** on Jester and **18.3%** on Epic-Kitchens), which highlights the great potential for improvement in future for unsupervised video domain adaptation.

**Comparison with MM-SADA [50].** MM-SADA[50] is another state-of-the-art approach for video domain adaptation that leverages the idea of using multi-modal (RGB and Optical flow) data to learn better domain invariant representations. The approach has two main components: adversarial learning and multi-modal supervision. While CoMix does not use optical flow features anywhere, the RGB-only version of MM-SADA still uses optical flow features for the multi-modal self-supervision. Interestingly, CoMix (43.2%) shows very competitive performance using only RGB features when compared to the above (43.9%) on the Epic-Kitchens dataset. Additionally, we train MM-SADA (RGB-only) (but perform multimodal supervision using both RGB and flow following the original paper [50]) on UCF-HMDB dataset and notice that CoMix outperforms it by a margin of 3% on an average (UCF → HMDB: 82.2% vs 86.7%, HMDB → UCF: 91.2% vs 93.9%, Avg: 86.7% vs 90.3%), showing its effectiveness in unsupervised video domain adaptation.

**Semi-supervised Domain Adaptation.** To further study the robustness of our proposed approach, we extend the unsupervised domain adaptation to a semi-supervised setting, where one (1-shot) and three target labels (3-shot) per class are available for training. Table 3 shows that our simple approach consistently outperforms the adversarial DA methods (DANN [20],

Table 3: **Semi-Supervised Domain Adaptation on UCF-HMDB and Jester Datasets.** CoMix significantly outperforms all the compared methods in both one-shot and three-shot settings.

| Method | UCF→HMDB | | HMDB→UCF | | Jester(S) → Jester(T) | |
|--------|----------|--------|----------|--------|--------|--------|
| | 1-shot | 3-shot | 1-shot | 3-shot | 1-shot | 3-shot |
| Source + Target | 83.2 | 85.8 | 90.3 | 93.7 | 53.8 | 55.0 |
| DANN [20] | 85.4 | 86.9 | 92.1 | 93.1 | 55.1 | 59.9 |
| ADDA [74] | 83.6 | 86.3 | 91.2 | 93.0 | 59.5 | 61.3 |
| ENT [63] | 85.6 | 88.6 | 92.8 | 95.8 | 58.6 | 61.5 |
| CoMix | **88.4** | **93.1** | **95.4** | **96.6** | **65.3** | **69.6** |

and ADDA [74]) including the semi-supervised method, ENT [63], on both UCF-HMDB and Jester datasets. Remarkably, CoMix with three target labels per class improves the performance of Source + Target baseline from **93.7%** to **96.6%**, which is only **0.2%** lower than the supervised target upper bound (in Table 1) on HMDB→UCF task (**96.6%** vs **96.8%**). These results well demonstrate the utility of our proposed approach in many practical applications where annotating *a few* videos per class is typically possible and therefore worth doing given the boost it provides.

**Effectiveness of Individual Components.** As seen from Table 4, the vanilla temporal contrastive learning (TCL) achieves an average accuracy of 85.8% on UCF-HMDB while 57.5% on Jester (1st row), which is already better than DANN [20], and ADDA [74] (ref. Table 1,2), showing its

effectiveness over adversarial learning in aligning features. While both background mixing (BGM) and incorporation of target pseudo-labels (TPL) individually improves the performance over TCL ($+2.9\%$, $+5.6\%$ using BGM and $+1.9\%$, $+5.4\%$ using TPL, respectively), addition of both of them leads to the best average performance of 90.3% on UCF-HMDB dataset and 64.7% on the Jester dataset. This corroborates the fact that both cross-domain action semantics (through BGM) and discriminabilty (through TPL) of the latent space play crucial roles in video domain adaptation in addition to the vanilla contrastive learning for aligning features.

Table 4: **Ablation Study on UCF-HMDB and Jester.** TCL: Temporal Contrastive Learning, BGM: Background Mixing, TPL: Target Pseudo-Labels.

| TCL | BGM | TPL | U→H | H→U | Average | Jester(S)→Jester(R) |
|---|---|---|---|---|---|---|
| ✓ | ✗ | ✗ | 83.3 | 88.4 | 85.8 | 57.5 |
| ✓ | ✓ | ✗ | 86.2 | 91.2 | 88.7 | 63.1 |
| ✓ | ✗ | ✓ | 83.5 | 91.9 | 87.7 | 62.9 |
| ✓ | ✓ | ✓ | **86.7** | **93.9** | **90.3** | **64.7** |

Table 5: **Comparison with MixUp Strategies.** Background mixing outperforms other alternatives in leveraging shared action semantics on UCF-HMDB.

| Method | U→H | H→U | Average | Jester(S)→Jester(R) |
|---|---|---|---|---|
| Gaussian Noise | 84.7 | 90.6 | 87.6 | 54.3 |
| Video MixUp [89] | 85.1 | 91.7 | 88.4 | 62.2 |
| Video CutMix [88] | 84.6 | 92.1 | 88.3 | 58.6 |
| Background Mixing | **86.7** | **93.9** | **90.3** | **64.7** |

**Comparison with Different MixUp Strategies.** We explore the effectiveness of background mixing by comparing with different MixUp strategies (Table 5): (a) Gaussian Noise: adding White Gaussian Noise to videos in both domains; (b) Video MixUp [89]: directly mixing one video with another from a different domain, as in images; (c) Video CutMix [88]: randomly replacing a region of a video with another region from the other domain. The proposed way of generating synthetic videos by mixing background of a video from one domain to a video from another domain, outperforms all three alternatives on UCF-HMDB as well as on the more challenging Jester dataset. Note that while both MixUp and CutMix destroy motion pattern of original video, background mixing keeps semantic consistency without changing the temporal dynamics.

**Effect of Background Pseudo-labels.** We investigate the effect of pseudo-labels on background mixed videos (i.e., both videos considered to be of same action class while creating positives) by simply adding them as unlabeled videos without any modification to the contrastive objective in Eq. 1. CoMix without background pseudo-labels decreases the performance from 90.3% to 89.0% ($-1.3\%$: Table 6), showing its effectiveness in leveraging action semantics shared across both domains.

**Effect of Source Contrastive Learning.** CoMix adopts contrastive learning on both source and target domains, although we already have supervised cross-entropy loss on source videos. We observe that applying contrastive learning on target domain only, by removing source contrastive objective $\mathcal{L}_{bgm}^{\{s\}}$ from Eq. 6, lowers down the performance from 90.3% to 88.4% ($-1.9\%$) on UCF-HMDB (Table 6). This shows the importance of training the model using the same temporal invariance objective on both domains simultaneously to achieve effective alignment across domains.

**Effect of Random Speed Invariance.** We remove randomness in video speed from the auxiliary branch of our temporal contrastive learning framework and observe that CoMix (with 16 clips in the base branch and only 8 clips in the auxiliary branch) leads to an average top-1 accuracy of 89.6% compared to 90.3% ($-0.7\%$:

Table 6: **Ablation Study on Contrastive Learning.**

| Method | U→H | H→U | Average |
|---|---|---|---|
| CoMix | **86.7** | **93.7** | **90.3** |
| – w/o Background Pseudo-labels | 85.8 | 92.2 | 89.0 |
| – w/o Source Contrastive Learning | 85.1 | 91.8 | 88.4 |
| – w/o Random Speed Invariance | 86.4 | 92.8 | 89.6 |

Table 6), showing the importance of random speed invariance in learning robust features.

**Self-Training vs Supervised Contrastive Learning.** We directly use self-training that uses cross-entropy loss on target pseudo labels instead of $\mathcal{L}_{tpl}^{\{t\}}$ and find that the average performance drops to 88.7% on UCF-HMDB, indicating the advantage of supervised contrastive objective in enhancing discriminability of the latent space by successfully leveraging label information from target domain.

**Effect of Graph Representation.** (a) *Removal of Graph Representation from* CoMix: We examine the effect of graph representation for videos and find that by removing GCN from our framework lowers down the performance from 90.3% to 88.1% on UCF-HMDB dataset, which shows that graph contrastive learning is more useful in

Table 7: **Baseline Comparisons w/ GCN Representations on UCF-HMDB and Jester Datasets.**

| Method (w/ GCN) | U→H | H→U | Average | Jester(S)→Jester(R) |
|---|---|---|---|---|
| Source Only | 82.5 | 87.7 | 85.1 | 54.0 |
| DANN [20] | 80.0 | 86.3 | 83.2 | 62.9 |
| TA³N [8] | 52.5 | 72.4 | 62.3 | 51.7 |
| CoMix | **86.7** | **93.9** | **90.3** | **64.7** |

capturing the temporal dependencies, essential for video domain adaptation. (b) *Effect of Graph*

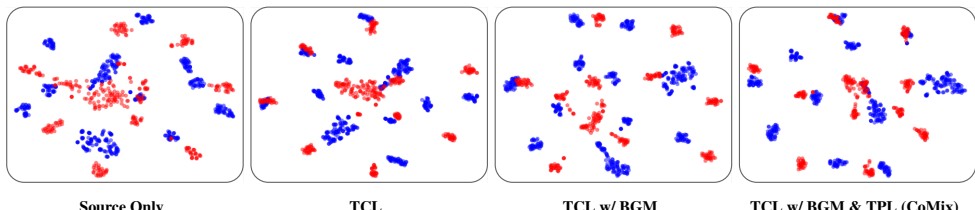

| Source Only | TCL | TCL w/ BGM | TCL w/ BGM & TPL (CoMix) |

Figure 5: **Feature Visualizations using t-SNE.** Plots show visualization of our approach with different components on UCF→HMDB task. Blue and red dots represent source and target data respectively. Features for both target and source domain become progressively discriminative and improve from left to right by adoption of our novel components within a temporal contrastive learning framework. Best viewed in color.

*Representation on Baseline Methods*: Additionally, in Table 7 we compare with domain adversarial adaptation methods DANN [20] and TA3N [8] including the Source only baseline with GCN feature representation on both UCF-HMDB and Jester datasets. CoMix improves the Source only accuracy by 5.2% and 10.7% respectively on UCF-HMDB and Jester datasets. Furthermore, CoMix outperforms DANN [20] with the same GCN equipped as ours, on both datasets (+7.1%, +1.8%, respectively) showing its effectiveness over adversarial learning in aligning features for video domain adaptation. TA3N [8] performs very poorly (62.3% and 51.7%) when equipped additionally with graph representations. We believe this is because TA3N already utilizes Temporal Relational Network [92] for modeling temporal relations, which probably hinders in learning GCN features for successful domain adaptation in videos. (c) *Alternatives for Graph Representation*: We replace GCN using MLP/LSTM of similar complexity and notice that both alternatives are inferior to GCN on UCF-HMDB (MLP: 88.1%, LSTM: 84.3%, GCN: 90.3%), which shows the effectiveness of GCN in our contrastive learning framework for capturing the temporal dependencies, essential for video domain adaptation.

**Effect of Background Extraction Method.** We experiment with a different background extraction strategy [93] that uses Gaussian Mixture Models (GMM) to extract the backgrounds and observe that the very simple and fast strategy based on temporal median filtering [58] outperforms GMM by 2.3% on average on UCF-HMDB (UCF→HMDB: 85.3% vs 86.7%, HMDB→UCF: 90.7% vs 93.9%, Avg: 88.0% vs 90.3%). Note that our CoMix framework is agnostic to the method used for background extraction and can be incorporated with any other background extraction techniques for videos, *e.g.*, learnable background segmentation strategies such as [82, 55].

**Feature Visualizations.** We use t-SNE [43] to visualize the features learned using different components of our CoMix framework. As seen from Figure 5, alignment of domains including discriminability improves as we adopt "TCL" and "BGM" to the vanilla Source only model. The best results are obtained when all the three components "TCL", "BGM" and "TPL" i.e., CoMix are added and trained using an unified framework (Eq. 6) for unsupervised video domain adaptation. Additional results and analysis including more qualitative examples are included in the supplementary material.

## 5 Conclusions

In this paper, we introduce a new end-to-end temporal contrastive learning framework to bridge the domain gap by learning consistent features representing two different speeds of the unlabeled videos. We also propose two novel extension to temporal contrastive loss by using background mixing and target pseudo-labels, that allows additional positive(s) per anchor, thus adapting contrastive learning to leverage cross-domain action semantics and label information from the target domain respectively in an unified framework, for learning discriminative invariant features. We demonstrate the effectiveness of our approach on three standard datasets, outperforming several competing methods.

**Broader Impact.** Our research can help reduce burden of collecting large-scale supervised data in many real-world applications of human action recognition by transferring knowledge from auxiliary datasets. The positive impact that our work could have on society is in making technology more accessible for institutions and individuals that do not have rich resources for collecting and annotating large-scale video datasets. Negative impacts of our research are difficult to predict, however, it shares many of the pitfalls associated with standard deep learning models such as susceptibility to adversarial attacks and lack of interpretablity. Other adverse effects could be potential attrition in jobs in certain sectors of economy where fewer employees (security guards, nurses, etc.) are needed to monitor human activities as a result of wider adoption of automated video recognition systems.

**Acknowledgements.** This work was partially supported by the ISIRD Grant EEE.

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
