# Contrast and Mix: Temporal Contrastive Video Domain Adaptation with Background Mixing (Supplementary Material)

**Aadarsh Sahoo**[1]    **Rutav Shah**[1]    **Rameswar Panda**[2]    **Kate Saenko**[2,3]    **Abir Das**[1]

[1] IIT Kharagpur, [2] MIT-IBM Watson AI Lab, [3] Boston University

{sahoo_aadarsh@, rutavms@, abir@cse.}iitkgp.ac.in, rpanda@ibm.com, saenko@bu.edu

Project Page: `https://cvir.github.io/projects/comix`

Table 1: **Overview of Supplementary Material.**

| Section | Content |
|---------|---------|
| A | Dataset Details |
| B | Temporal Graph Encoder |
| C | Additional Implementation Details |
| D | Background Extraction Details |
| E | Additional Experimental Results |
| F | Additional Feature Visualizations |

## A    Dataset Details

In this section, we provide the detailed description of the datasets we used to perform all the experiments for `CoMix`, namely, (1) UCF-HMDB [3], (2) Jester [13], and (3) Epic-Kitchens [12].

**UCF-HMBD Dataset**. The UCF-HMDB dataset (assembled by [3]) is derived from the original UCF101 [15] and HMDB51 [7]. It is constructed by collecting all the relevant and overlapping action classes or categories from both the datasets as two domains, resulting in 2 transfer tasks (UCF→HMDB and HMDB→UCF). The dataset possesses 12 action classes, namely, *Climb, Fencing, Golf, Kick_Ball, Pullup, Punch, Pushup, Ride_Bike, Ride_Horse, Shoot_Ball, Shoot_Bow,* and *Walk*. For some of the cases, multiple action classes from the original dataset are combined to form a single action super-class for that domain. E.g., *RockClimbingIndoor* and *RopeClimbing* classes in the HMDB51 [7] dataset are combined to form *Climb* class for the HMDB domain in the UCF-HMDB dataset. The detailed composition of the action classes is shown in Table 2. The dataset contains $3,209$ videos in total with $1438$ training videos and $571$ validation videos from UCF, and $840$ training videos and $360$ validation videos from HMDB (following the splits by [3]), with a class-wise distribution shown in Figure 1.

The datasets are publicly available to download at:
`https://www.crcv.ucf.edu/data/UCF101.php`
`https://serre-lab.clps.brown.edu/resource/hmdb-a-large-human-motion-database/`.

**Jester Dataset.** The Jester [11] dataset is a large scale fine-grained dataset consisting of videos of humans performing pre-defined hand gestures. The original dataset consists of $148092$ videos from $27$ action classes. A cross-domain dataset is constructed (originally by [13]) as a subset of the original dataset by merging multiple action classes into a single action super-class and then split into source and target domain. E.g., *Swiping Left, Swiping Right, Swiping Up,* and *Swiping Down* are considered

Table 2: **Action classes in UCF-HMDB Dataset.** The table shows the action class composition for the UCF-HMDB dataset from the original datasets (i.e., UCF101 and HMDB51) and their correspondence to each other for the video domain adaptation setting.

| UCF-HMDB | HMDB51 | UCF101 |
|:---:|:---:|:---:|
| Climb | climb | RockClimbingIndoor RopeClimbing |
| Fencing | fencing | Fencing |
| Golf | golf | GolfSwing |
| Kick_Ball | kick_ball | SoccerPenalty |
| Pullup | pullup | PullUps |
| Punch | punch | Punch |
| Pushup | pushup | PushUps |
| Ride_Bike | ride_bike | Biking |
| Ride_Horse | ride_horse | HorseRiding |
| Shoot_Ball | shoot_ball | Basketball |
| Shoot_Bow | shoot_bow | Archery |
| Walk | walk | WalkingWithDog |

as a super-class *Swiping*. Then *Swiping Left, Swiping Up* are considered to be in the source domain, while *Swiping Right, Swiping Down* to be in the target domain. Different sub-actions are put into different domains in order to maximize the domain discrepancy, as stated by [13]. The resulting cross-domain dataset possesses 7 action classes, namely, *Push and Pull, Rolling Hand, Sliding Two Fingers, Swiping, Thumps Up and Down, Turning Hand,* and *Zooming In and Out.* For Jester, we have only a single transfer task i.e. Jester(S)→Jester(T). The detailed composition of the action classes is shown in Table 3 with a class-wise distribution of videos depicted in Figure 2.

The dataset is publicly available to download at: `https://20bn.com/datasets/jester`.

Table 3: **Action Classes in Jester.** The table shows the action class composition for each of the domains (i.e. Source and Target) in the Jester dataset and their correspondence to each other for the domain adaptation setting.

| Jester | Jester (S) | Jester (T) |
|:---:|:---:|:---:|
| Push and Pull | Pushing Hand Away Pushing Two Fingers Away | Pulling Hand In Pulling Two Fingers In |
| Rolling Hand | Rolling Hand Forward | Rolling Hand Backward |
| Sliding Two Fingers | Sliding Two Fingers Left Sliding Two Fingers Up | Sliding Two Fingers Right Sliding Two Fingers Down |
| Swiping | Swiping Left Swiping Up | Swiping Right Swiping Down |
| Thumps Up and Down | Thumb Up | Thumb Down |
| Turning Hand | Turning Hand Counterclockwise | Turning Hand Clockwise |
| Zooming In and Out | Zooming Out With Full Hand Zooming Out With Two Fingers | Zooming In With Full Hand Zooming In With Two Fingers |

**Epic Kitchens Dataset.** The Epic-Kitchens [5] dataset is a challenging egocentric dataset consisting of videos (action segments) capturing daily activities performed in kitchens. The three largest kitchens, namely, P01, P22, and P08 form the three domains D1, D2, and D3, respectively. Moreover, the 8 largest action classes, namely, *take, put, open, wash, close, cut, pour,* and *mix* are used to form the dataset for the domain adaptation setting, following [12]. The dataset has 1543 training videos and 435 test videos from D1, 2495 training videos and 750 test videos from D2, and 3897 training videos and 974 test videos from D3. The class-wise distribution is shown in Figure 3. It can be seen that the dataset possesses high imbalance which makes it even more challenging.

The dataset is publicly available to download at: `https://epic-kitchens.github.io/2021`.

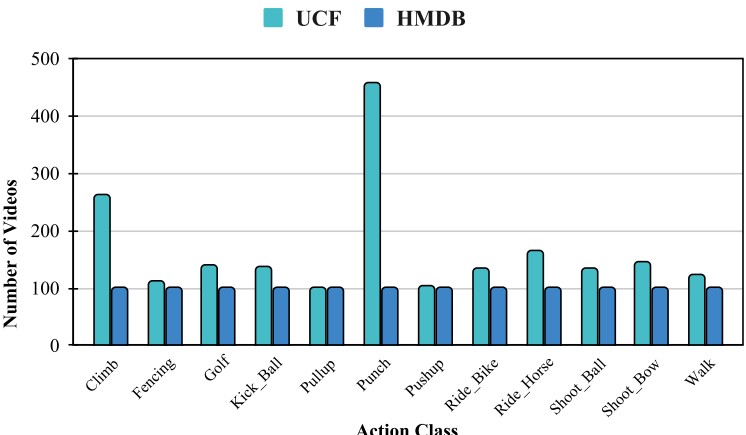

Figure 1: **Class-wise distribution of videos for UCF-HMDB.** The bar chart shows the distribution of videos across the 12 action classes of the UCF-HMDB dataset. Best viewed in color with zoom.

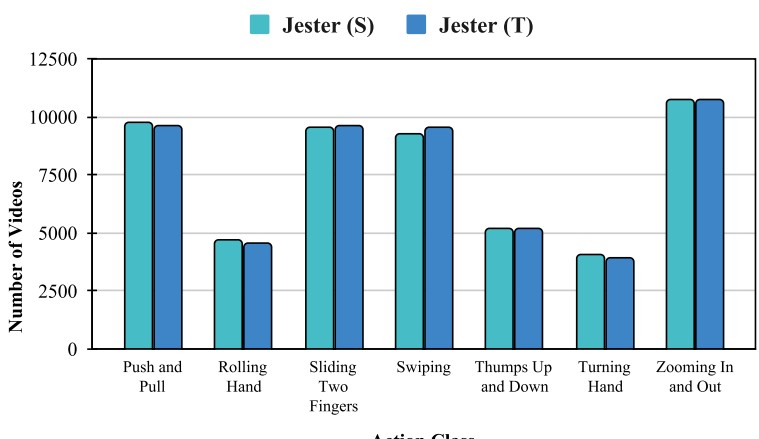

Figure 2: **Class-wise distribution of videos for Jester.** The bar chart shows the distribution of videos across the 7 action classes of the Jester dataset for the source and the target domains. Best viewed in color with zoom.

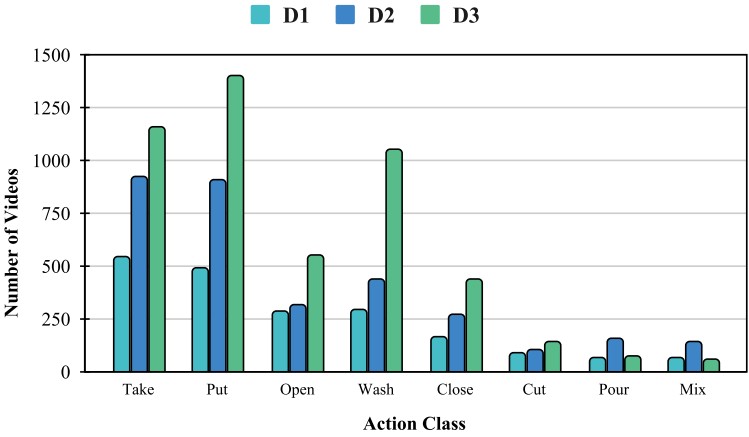

Figure 3: **Class-wise distribution of videos for Epic-Kitchens.** The bar chart shows distribution of videos across 8 action classes of Epic-Kicthens for three domains D1, D2, and D3. Best viewed in color with zoom.

# B  Temporal Graph Encoder

In this section, we provide the detailed description of the temporal graph encoder that we used for representing videos in our contrastive learning framework.

## B.1  Graph Convolutional Network

The graph convolutional network (GCN) was originally proposed by [6] for node classification on graph structured data. Given an input graph $X \in \mathbb{R}^{N \times d}$ with $N$ number of nodes with each node as a feature-vector of dimension $d$, the layer-wise propagation rule for a multi-layer GCN is:

$$H^{(l+1)} = \sigma(\tilde{D}^{-\frac{1}{2}} \tilde{A} \tilde{D}^{-\frac{1}{2}} H^{(l)} W^{(l)}) \tag{1}$$

where, $H^{(l)} \in \mathbb{R}^{N \times d_l}$ is the activation graph of the $l^{\text{th}}$ layer with node feature dimension $d_l$; $H^{(0)} = X$. $\tilde{A} = A + I_N$ is the adjacency matrix of $X$ with added self-connections through the indentity matrix $I_N$. $\tilde{D}_{ii} = \sum_j \tilde{A}_{ij}$ is the diagonal matrix used for normalization of $\tilde{A}$, and $W^{(l)}$ is the layer-specific trainable weight matrix. $\sigma(.)$ denotes the activation function, e.g. $\text{ReLU}(.)$.

## B.2  Videos as Similarity Graphs

Motivated by the importance of capturing long-range temporal structure in videos for action recognition and hence in cross-domain adaptation, we adopt a similarity graph to represent a video in our framework, as in [17]. Given a video $\mathbf{V}_n = \{\mathbf{v}_1, \mathbf{v}_2, ..., \mathbf{v}_n\}$ with $n$ clips, with the corresponding clip-level feature vector representations as $\mathbf{Z}_n = \{\mathbf{z}_1, \mathbf{z}_2, ..., \mathbf{z}_n\}$, extracted by the feature encoder $\mathcal{F}$, each of dimension $d$. We construct a fully-connected graph $X$ with $n$ nodes from $\mathbf{Z}$ by considering the pairwise similarity or affinity between two feature vectors as:

$$F(\mathbf{z}_i, \mathbf{z}_j) = \phi(\mathbf{z}_i)^\top \phi'(\mathbf{z}_j) \tag{2}$$

where, $\phi(.)$ and $\phi'(.)$ represent two different transformation functions of the original feature vectors, defined as $\phi(\mathbf{z}) = \mathbf{wz}$ and $\phi(\mathbf{z}) = \mathbf{w}'\mathbf{z}$. Here, the transformations are parameterized with the weights $\mathbf{w}$ and $\mathbf{w}'$ of dimension $d \times d$ each. Using such transformations helps learn the long-range correlations between the feature vectors to harness the rich temporal information of the video. We get a similarity matrix $A^{sim}$ of dimension $n \times n$ by computing the affinity for all the possible pairs, using Eq. 2. The matrix is then normalized using a softmax function as:

$$A^{sim}_{ij} = \frac{\exp(F(\mathbf{z}_i, \mathbf{z}_j))}{\sum\limits_{j=1}^{n} \exp(F(\mathbf{z}_i, \mathbf{z}_j))} \tag{3}$$

The normalized matrix $A^{sim}$ is now considered as the adjacency matrix for the similarity graph, allowing us to learn the edge-weights between the nodes through back-propagation, by the help of the learnable weights $\mathbf{w}$ and $\mathbf{w}'$. Hence, the resulting similarity graph convolutional network has the following propagation rule, similar to Eq. 1:

$$H^{(l+1)} = \sigma(A^{sim(l)} H^{(l)} W^{(l)}) \tag{4}$$

where, $H^{(0)} = X$, and $A^{sim(l)}$ is the affinity/adjacency matrix computed using the node features of the $l^{\text{th}}$ layer, similar to [17].

## B.3  Scalability with Graph Convolutions

The learning strategy for graph representation used in CoMix ensures that the number of learnable parameters are independent of the number of graph nodes. As described in detail above, we construct the fully connected graph with the edge weights (pairwise similarity) obtained using two different transformation functions, and on the clip-level feature vectors (where each feature vector represents a node). This strategy makes the number of trainable parameters independent of the number of nodes in a GCN layer and hence, independent of the number of clips used for a video. While fully connected graph convolutions will increase the computation with longer clip sequences, we can adopt sparse video sampling [2] or techniques like [18] to tradeoff computation.

# C  Additional Implementation Details

In this section, we provide additional implementation details including hyperparameters with a detailed overview of the model architectures used in our approach.

## C.1  Model Architectures

**Feature Encoder.** Following [4], we use I3D [1] as our feature encoder $\mathcal{F}$ for all our experiments. It takes clips (set of consecutive frames) of videos, of length 8, as input and maps them to the corresponding clip-level feature vector of length 1024. The layer-wise architectural view of the I3D feature encoder backbone is shown below:

```
InceptionI3D:
  (Conv3d_1a_7x7): Unit3D()
  (MaxPool3d_2a_3x3): MaxPool3dSamePadding()
  (Conv3d_2b_1x1): Unit3D()
  (Conv3d_2c_3x3): Unit3D()
  (MaxPool3d_3a_3x3): MaxPool3dSamePadding()
  (Mixed_3b): InceptionModule()
  (Mixed_3c): InceptionModule()
  (MaxPool3d_4a_3x3): MaxPool3dSamePadding()
  (Mixed_4b): InceptionModule()
  (Mixed_4c): InceptionModule()
  (Mixed_4d): InceptionModule()
  (Mixed_4e): InceptionModule()
  (Mixed_4f): InceptionModule()
  (MaxPool3d_5a_2x2): MaxPool3dSamePadding()
  (Mixed_5b): InceptionModule()
  (Mixed_5c): InceptionModule()
  (avg_pool): AvgPool3d()
  # Outputs features of length 1024.
```

**Temporal Graph Encoder** For the temporal graph encoder $\mathcal{G}$, we use a 3-layer similarity based graph convolutional neural network, as discussed in Section B. The graph encoder takes the output of the feature encoder $\mathcal{F}$ as input and gives the logits as the output. The layer-wise architectural view of the temporal graph encoder is shown below:

```
TemporalGraph:
  # Takes the output of InceptionI3d as input.
  (gc1): GraphConvolution(1024, 256)
  (relu): ReLU()
  (dropout): Dropout()
  (gc2): GraphConvolution(256, 256)
  (relu): ReLU()
  (dropout): Dropout()
  (gc3): GraphConvolution(256, num_classes)
  # Outputs the logits.
```

## C.2  Hyperparameters

Below, we provide the exact values of the two loss weights $\lambda_{bgm}$ and $\lambda_{tpl}$ (refer to Eq. 6 in the main paper) for each of the datasets:

**UCF-HMDB**: UCF→HMDB: $\lambda_{bgm} = 0.1$, $\lambda_{tpl} = 0.01$; HMDB→UCF: $\lambda_{bgm} = 0.1$, $\lambda_{tpl} = 0.1$.

**Jester**: S→T: $\lambda_{bgm} = 0.1$, $\lambda_{tpl} = 0.1$.

**Epic-Kitchens**: $\lambda_{bgm} = 0.01$, $\lambda_{tpl} = 0.01$ was used for all the 6 tasks.

For all the experiments, the source-only models were trained for 4000 iterations and then our framework was trained for an additional 10000 iterations, initialized with the source-only models.

## D    Background Extraction Details

In this section, we provide more details about the background extraction including qualitative samples used in our temporal contrastive learning framework.

**Temporal Median Filter.** Temporal median filtering (TMF) is one of the most simple, intuitive, and fast methods for background generation. It has proven to be successful and commonly adopted in several recent deep learning pipelines [16, 10]. For videos, a pixel-wise temporal median filter is applied on the sequence of frames to obtain the corresponding background. The method is designed with the principle that for a given pixel location, in a sequence of frames, the most frequently repeated intensity along the temporal direction is most likely to be the background value for that pixel [14, 8]. It does so by computing the pixel-wise median values along the temporal direction. We adopt this method for extracting backgrounds for our framework because of its simplicity and effectiveness. It must be noted that the `CoMix` framework is agnostic to the method used for background extraction and can be incorporated with any other background extraction techniques for videos. Figure 4 shows some representative video clips randomly sampled from both the domains of the UCF-HMDB along with the corresponding background frame extracted using temporal median filtering. Note that we extract a single background frame per video from one domain and then mix it with all the frames of a video from the other domain to generate synthetic background mixed videos. The addition of a static background frame to all frames of a video does not hinder the temporal action dynamics (motion patterns) possessed by the video. We validate this hypothesis by obtaining optical flow for a given video before and after performing background mixing, and observe no significant change in them.

## E    Additional Experimental Results

**Effect of Source-only Model Initialization.** We follow the standard 'pre-train then adapt' procedure used in prior works [3, 4] and train the model with only source data to provide a warmstart before our approach is trained. However, in order to understand the contribution of source-only model initialization, we trained the models with the default random initialization keeping all the other hyperparameters same. The average performance dropped to $86.4\%$ (-**3.9**%) on UCF-HMDB dataset. This validates that the source-only initialization plays an important role in providing a proper warmstart to the models which leads to an effective optimization, in consistent with prior works [3, 4].

**Effect of Target Pseudo-label Threshold.** In Table 4, we study the sensitivity of the final performance with respect to the pseudo label threshold on the UCF-HMDB dataset and notice that the performance of `CoMix` is quite stable with respect to this parameter (best performance at threshold set to $0.7$). The slight decrease in performance with PL = $0.9$ is understandable since very few target videos are getting selected as additional positives for the supervised contrastive loss.

**Effect of Mixed Backgrounds.** We tried a variant of background mixing in which the backgrounds from both the domains are first convexly combined to form a mixed-background, which sort of represents a generalized background for both the domains. The obtained mixed-background is then used for the background mixing component and is mixed with the videos from both the domains. This alternate background mixing strategy provided an average performance of $89.4\%$ on the UCF-HMDB dataset, which is $0.9\%$ lower than the cross-domain background mixing (i.e., adding background from one domain to the other) used in the proposed approach.

Table 4: **Effect of Target Pseudo-label Threshold.** Performance on UCF-HMDB.

| PL Threshold | U→H | H→U | Average |
|---|---|---|---|
| 0.5 | 85.6 | 93.5 | 89.5 |
| 0.6 | 85.6 | 93.5 | 89.5 |
| 0.7 | **86.7** | **93.9** | **90.3** |
| 0.8 | 86.4 | 92.5 | 89.4 |
| 0.9 | 85.6 | 90.9 | 88.2 |

**Convergence and Multiple Seeds.** The convergence of the proposed approach varies with dataset and task complexity ranging from 3000 iterations for HMDB→UCF dataset to 7000 iterations for UCF→HMDB and EpicKitchens datasets. We observe that the convergence is fairly stable across different seeds and report the average performance over three runs with different random seeds. To quantify, the standard deviations in performance obtained for UCF-HMDB, Jester and EpicKitchens datasets are 0.3, 0.1 and 0.2 respectively.

| Original Clip | | | | Background using TMF |
|---|---|---|---|---|
| **Sample Clips from the UCF domain of UCF-HMDB dataset** | | | | |

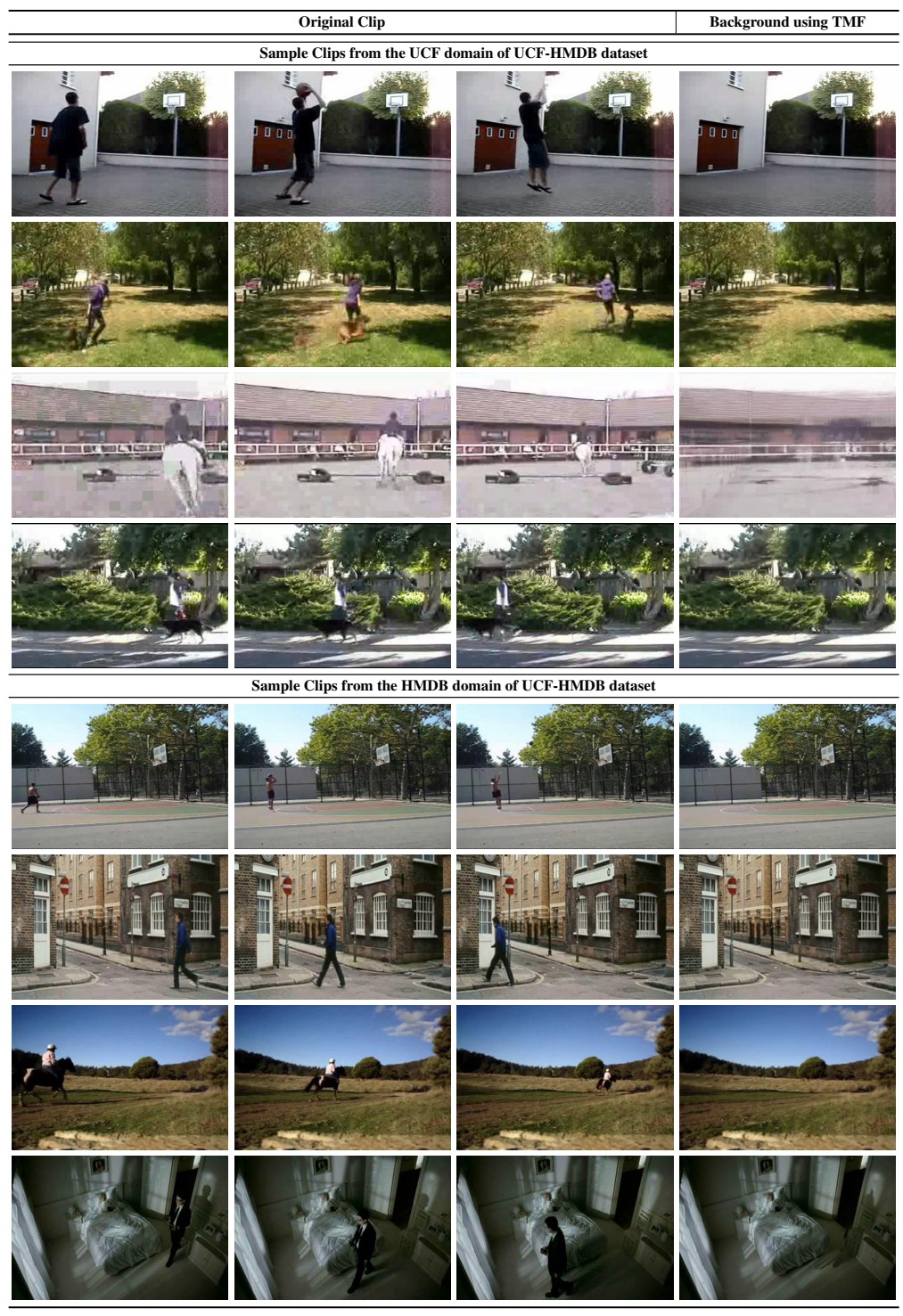

**Sample Clips from the HMDB domain of UCF-HMDB dataset**

Figure 4: **Background Extraction.** The figure shows some representative video clips from UCF-HMDB dataset with corresponding extracted background using temporal median filtering (TMF). Best viewed in color.

## F Additional Feature Visualizations

In this section, we provide additional t-SNE [9] plots to visualize the features learned using different components of our CoMix framework. We choose the Source only model as a vanilla method and add different components one-by-one to visualize their contributions in learning discriminative features

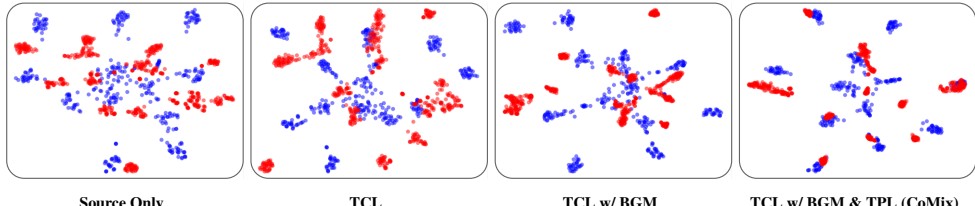

| Source Only | TCL | TCL w/ BGM | TCL w/ BGM & TPL (CoMix) |

Figure 5: **Feature Visualizations using t-SNE.** Plots show visualization of our approach with different components on HMDB→UCF task from UCF-HMDB. Blue and red dots represent source and target data respectively. Features for both target and source domain become progressively discriminative and improve from left to right by adoption of our novel components within a contrastive learning framework. Best viewed in color.

for video domain adaptation. In the main paper, we have provided the plots for the UCF→HMDB task from the UCF-HMDB dataset (refer to Figure 5 in main paper). Here we provide the plots for the HMDB→UCF task in Figure 5. As can be seen from Figure 5, alignment of domains including discriminability improves as we adopt "TCL" and "BGM" to the vanilla Source only model. The best results are obtained when all three components "TCL", "BGM" and "TPL" i.e., CoMix are added and trained using an unified framework (Eq. 6 in main paper) for unsupervised video domain adaptation.