# OpenReview forum: "Contrast and Mix: Temporal Contrastive Video Domain Adaptation with Background Mixing"
_NeurIPS.cc/2021/Conference — NeurIPS 2021 Poster_

### Official Review · Reviewer_CMgF · 2021-07-13

**Rating:** 6
**Confidence:** 4

**Summary:**

In this paper, the authors tackle the problem of video domain adaptation for the task of action recognition. To align source and target domain features, they propose to use contrastive learning instead of domain adversarial learning. For contrastive learning, two instances of different speeds from the same video constitute a positive pair. Two instances of different speed from the different video constitute a negative pair. In addition to the contrastive learning, they propose the background mixup in order to bridge the domain gap between source background and target background. They also incorporate target pseudo-labels to learn more discriminative target classifier. They validate the efficacy of the model on the publicly available benchmarks: UCF-HMDB, Jester and Epic-Kitchens.

The claimed contribution of the paper are as follows.
 1) Introducing a new UDA framework, Contrast and Mix (CoMix): contrastive learning with speed, and background mixup for bridging the domain gap
2) Integrate the pseudo labeling for UDA
3) Extensive experiments on three datasets, and achieving state-of-the-art performance on UCF-HMDB and Jester.


**Limitations And Societal Impact:**

Yes.

**Main Review:**

This paper has the following strengths.

+ The problem addressed, unsupervised domain adaptation for video recognition, is an interesting, but unsolved problem.
+ The proposed method has a good motivation and sensible design choices. Speed is an almost invariant feature for recognizing actions in many cases (except a few examples such as walk vs. jog/run, slow dance vs. fast dance). Background shift between source videos and target videos is a known problem. Pseudo-labeling is shown to be helpful for the UDA in image domain [1].
+ The paper is structured well. It is easy for readers to follow except for a few confusing sentences.

[1] Unsupervised Domain Adaptation for Distance Metric Learning, Sohn et al., ICLR 2019.


However, this paper has weaknesses as well.
- The comparison with adversarial domain adaptation methods (ADDA, DANN, TA$^3$N, SAVA or even the source only baseline) is unfair in Table 1. CoMix is based on the video-level representations from GCN, while the compared methods are not using GCN representations. The authors claim the proposed method is more effective than domain adversarial methods based on this result. However, we do not know where the improvement comes from. Does it come from GCN or CoMix (contrastive learning with background mixup)? The ablation on the GCN shows CoMix w/o GCN gives 88.1% on the UCF-HMDB which is close to SAVA (86.7%, note that SAVA does not have pseudo-labeling) than CoMix with GCN. Therefore, it is not straightforward to draw conclusion that CoMix is better than adversarial DA methods. It would be better to see the results from the source only, DANN and ADDA baselines with GCN equipped.
- There is no comparison with MM-SADA [51] on the Epic-Kitchens dataset. MM-SADA is a multi-modal method. Therefore, it is unfair to compare CoMix (43.2% on avg) with RGB+Flow version of MM-SADA (50.3% on avg). However, comparison with RGB only MM-SADA (43.9% on avg) should be there. RGB only MM-SADA (43.9%) shows slightly higher performance than CoMix (43.2%). This result might suggest adversarial learning is still valuable for the domain adaptation problem.
- All ablation experiments are conducted on UCF-HMDB which is a quite small (only 3K videos with 12 classes), and performance-saturated dataset. It is hard to draw concrete conclusions from UCF-HMDB experiments. It is better to conduct ablation experiments on the Epic-Kitchens or Jester dataset which are larger and less performance-saturated.

If the majority of my concerns are resolved by the rebuttal, I will consider increasing my rating.


**Time Spent Reviewing:**

5 hours

---

> ### Author Response · Authors · 2021-08-10
> **Response to Reviewer CMgF**
>
> We appreciate the reviewer for acknowledging that the proposed method has good motivation and sensible design choices for addressing the problem of unsupervised domain adaptation for video recognition! Below we address the concerns regarding baseline comparison with GCN equipped and additional ablation experiments on the Jester dataset.
>
> (a) **Baseline Comparisons w/ GCN Representations**: Following reviewer’s suggestion, We compare with domain adversarial adaptation methods DANN and TA3N including the Source only baseline with GCN feature representation on both UCF-HMDB and Jester datasets. As can be seen from the table below, our proposed CoMix improves the Source only accuracy by 5.2% and 10.7% respectively on UCF-HMDB and Jester datasets. Furthermore, CoMix clearly outperforms DANN with the same GCN equipped as ours, on both datasets showing its effectiveness over adversarial learning in aligning features for video domain adaptation. Interestingly, TA3N with GCN representation fails to outperform the Source only baseline on both datasets. We believe this is because TA3N already utilizes Temporal Relational Network (TRN) for modeling temporal relations, which probably hinders in learning the GCN features for successful domain adaptation in videos. Note that we were not able to compare SAVA using GCN features as the code is not yet publicly available.
>
> \\begin{array} {|lcccc|}
> \\hline \\textbf{Methods (w/ GCN)} & \\textbf{U &#8594; H} & \\textbf{H &#8594; U} & \\textbf{Average} & \\textbf{Jester(S) &#8594; Jester(T)} \\\\
> \\hline
> \\text{Source Only}  & \\text{82.5} & \\text{87.7} & \\text{85.1} & \\text{54.0} \\\\
> \\text{DANN}  & \\text{80.0} & \\text{86.3} & \\text{83.2} & \\text{62.9} \\\\
> \\text{TA3N}  & \\text{52.5} & \\text{72.4} & \\text{62.3} & \\text{51.7} \\\\
> \\text{CoMix (Ours)}  & \\textbf{86.7} & \\textbf{93.9} & \\textbf{90.3} & \\textbf{64.7} \\\\
> \\hline
> \\end{array}
>
> (b) **Comparison with MM-SADA [1]**: Thank you for pointing out the missing clarification on MM-SADA results. MM-SADA leverages the idea of using multi-modal (RGB and Optical flow) data to learn better domain invariant representations, while our proposed approach only utilizes RGB frames in a contrastive learning framework. More specifically, MM-SADA has two main components such as adversarial learning and multi-modal supervision. As rightly said by the reviewer, it is unfair to compare CoMix with RGB+Flow version of MM-SADA. However, the MM-SADA (RGB only) results provided by the authors still uses flow for the self-supervised component, as can be seen from Table 3 caption in [1]. Interestingly, our CoMix (43.2%) shows very competitive performance while using only RGB when compared to MM-SADA (RGB) (43.9%) that still utilizes flow on the Epic-Kitchens dataset. Additionally, we also compare with MM-SADA (RGB) (trained with multimodal supervision using RGB and flow similar to the original paper [1]) on UCF-HMDB dataset and notice that CoMix outperforms it by a margin of 3% on an average (UCF→HMDB: 82.2% vs 86.7%, HMDB→UCF:91.2% vs 93.9%, Avg: 86.7% vs 90.3%), showing its effectiveness in unsupervised video domain adaptation. We will add a detailed discussion on comparison with MM-SADA in the final version.
>
>
> (c) **Ablation Experiments on Jester Dataset**: Thanks for the suggestion!
>
> - **Effectiveness of Individual Components**: We report this ablation study in Section E of the supplementary material (Table 4). As can be seen from the table below, the vanilla temporal contrastive loss (TCL) achieves an average accuracy of 57.5% (1st row), reiterating its effectiveness over adversarial learning for domain alignment. Adding background mixing (BGM) and target pseudo-labels (TPL) improves the performance over TCL by +5.6% and +5.4% respectively. Adding both of them to the framework gives an average performance of 64.7% on the Jester dataset. This re-validates the fact that both cross-domain action semantics (through BGM) and discriminability (through TPL) of the latent space plays a crucial role in video domain adaptation in addition to vanilla contrastive learning for aligning features.
>
> \\begin{array} {|cccc|}
> \\hline \\textbf{TCL} & \\textbf{BGM} & \\textbf{TPL} & \\textbf{Jester(S) &#8594; Jester(T)} \\\\
> \\hline
> &check; & &cross; & &cross; & \\text{57.5} \\\\
> &check;  & &check; & &cross; & \\text{63.1} \\\\
> &check;  & &cross; & &check; & \\text{62.9} \\\\
> &check;  & &check; & &check; & \\textbf{64.7} \\\\
> \\hline
> \\end{array}
>
> - **Comparison with Different MixUp Strategies**: As seen from the table below, our proposed way of generating synthetic videos by mixing the background of a video from one domain to a video from another domain clearly outperforms all the three alternatives (Gaussian Noise, Video MixUp, and Video CutMix) on the Jester dataset. This is consistent with the results on UCF-HMDB (Table 5 of the main paper), which shows that our proposed background mixing keeps semantic consistency without changing temporal dynamics leading to an improvement in performance on cross-domain video domain adaptation.
>
> \\begin{array} {|cc|}
> \\hline \\textbf{Method} & \\textbf{Jester(S) &#8594; Jester(T)} \\\\
> \\hline
> \\text{Gaussian Noise} & \\text{54.3} \\\\
> \\text{Video MixUp} & \\text{62.2} \\\\
> \\text{Video CutMix} & \\text{58.6} \\\\
> \\text{Background Mixing (Ours)} & \\textbf{64.7} \\\\
> \\hline
> \\end{array}
>
> - **Other Ablation Experiments**: We also test the effectiveness of other components like source contrastive learning, random speed invariance, self-training vs supervised contrastive learning on Jester dataset and notice a similar trend as UCF-HMDB dataset. We will add all these results in the final version.

---

> > ### Comment · Reviewer_CMgF · 2021-08-23
> > **Increasing my rating**
> >
> > I have read the rebuttal and other reviews. The majority of my concerns are resolved by the rebuttal. The rebuttal shows that the proposed method has its merit by exploiting background mixup, and temporal contrastive learning with speed augmentations. I am increasing my rating.

---

> > > ### Author Response · Authors · 2021-08-24
> > > **Thanks**
> > >
> > > Thank you for your response.  We're glad that you found our temporal contrastive learning with background mixing has significant merits.

---

### Official Review · Reviewer_bY6e · 2021-07-15

**Rating:** 7
**Confidence:** 3

**Summary:**

This paper proposes a new contrastive method for unsupervised video domain adaptation. This method employs temporal contrastive learning based on graph convolution and back ground mixing approach for enhancing action similarity, so called CoMix.
Additionally, for tagert domain, CoMix uses a pseudo label-based target domain contrastive learning. The authors evaluate CoMix on UFC-HMDB, Epic-Kitchen, and Jester datasets with intensive comparison and ablation studies.

**Ethical Concerns:**

No ethical concern

**Limitations And Societal Impact:**

#### No negative societal impact


**Main Review:**

### Strength
- Unsupervised video domain adaption is a challenging but significant research topic.
- Leveraging contrastive learning, graph representation for temporal relations, and background mixing in video domain adaption is an interesting idea and seems novel (considering NeurIPS deadline).
- It is well-written, well-organized, and easy to read.
- Including ablation study, experiments seem thorough except for the results on Epic-Kitchen dataset.
- Source code was submitted.

### Weakness
- Some important previous references were missed in related work such as
  - For domain adaption: MCD [Saito et al. 2018]
  - For Image Mixture: PuzzleMix [Kim et al. 2020] and CoMixup [Kim et al. 2021]
- Also, the comparison with MM-SADA [Munro and Damen 2020] was missed even if it was referred to in the paper as [51]. In particular, the comparison is important because MM-SADA proposes Epic-Kitchen dataset and MM-SADA with RGB only not using optical flow shows comparable or better performances than the proposed CoMix. This should be clarified.
- The authors employs graph convolution for temporal contrastive learning. In this study, because the number of clips are not large (=16), the fully connected graph is not a problem. However, when the number of clips increases (for longer clip sequence), the connections also exponentially increases, which might restrict scale-up of this method.
-  For background mixing, the authors uses a very simple image processing method. How about using more advanced pretrained foreground-background decomposition method? How large can the background mixing method influence on the performance?
- How sensitive are the results by target pseudo-label confidence threshold? Any results?
- For GCN ablation, did the author try to use MLP, LSTM, or Transformer instead of GCN? Because removing GCN makes the total parameter reduced, other NN can be helpful to validate the effect of GCN.
- Minor comments
  - L276 and L308 in P7, SAVA was wrongly cited. [10] --> [12]
  - Overall, there is some redundancy in Introduction, Proposed Method and captions in Figures 2 and 3.
  - Figure 3 can be more detailed. If the page limitation is an issue, More detailed figure can be located at supplementary material.
  - Instead of naive video cutmix, VideoMix [Yun et al. 2020] can be used.
  - [Song et al. 2021] can be added in related work although it is not published until the NeurIPS deadline.

#### References
- [Saito et al. 2018] Maximum classifier discrepancy for unsupervised domain adaptation. CVPR 2018.
- [Kim et al. 2020] Puzzle Mix: Exploiting Saliency and Local Statistics for Optimal Mixup. ICML 2020.
- [Kim et al. 2021] Co-Mixup: Saliency Guided Joint Mixup with Supermodular Diversity. ICLR 2021.
- [Yun et al. 2020] VideoMix: Rethinking Data Augmentation for Video Classification. arXiv 2020.
- [Song et al. 2021] Spatio-temporal Contrastive Domain Adaptation for Action Recognition. CVPR 2021.


=========================== After rebuttal =======================
I read the other reviewers' comments and author response. Beacuse the authors addressed most of my concerns, I decided to raise my score.


**Time Spent Reviewing:**

10

---

> ### Author Response · Authors · 2021-08-10
> **Response to Reviewer bY6e**
>
> We thank the reviewer for acknowledging that our paper is addressing a challenging yet significant research topic, and that leveraging contrastive learning is an interesting novel idea in video domain adaptation.
>
> (a) **Comparison with MM-SADA [1]**: Thank you for pointing out the missing clarification on MM-SADA results. MM-SADA leverages the idea of using multi-modal (RGB and Optical flow) data to learn better domain invariant representations, while our proposed approach only utilizes RGB frames in a contrastive learning framework. More specifically, MM-SADA has two main components such as adversarial learning and multi-modal supervision. The MM-SADA (RGB only) results provided by the authors still uses flow for the self-supervised component, as can be seen from Table 3 caption in [1]. Interestingly, our CoMix (43.2%) shows very competitive performance while using only RGB when compared to MM-SADA (RGB) (43.9%) that still utilizes flow on the Epic-Kitchens dataset. Additionally, we also compare with MM-SADA (RGB) (trained with multimodal supervision using RGB and Flow similar to the original paper [1]) on UCF-HMDB dataset and notice that CoMix outperforms it by a margin of 3% on an average (UCF→HMDB: 82.2% vs 86.7%, HMDB→UCF:91.2% vs 93.9%, Avg: 86.7% vs 90.3%), showing its effectiveness in unsupervised video domain adaptation. We will add a detailed discussion on comparison with MM-SADA in the final version.
>
> (b) **Scalability with Graph Convolutions**: The learning strategy used in CoMix ensures that the number of learnable parameters are independent of the number of graph nodes. Specifically, we construct the fully connected graph with the edge weights (pairwise similarity) obtained using two different transformation functions, $\phi(.)$ and $\phi^\prime(.)$ on the clip-level feature vectors (where each feature vector represents a node). The transformation functions are parameterized using two learnable weight matrices, $\mathbf{w}$ and $\mathbf{w}^\prime$, of dimension $d_{in} \times d_{in}$, with $d_{in}$ being the dimension of the input feature vectors. Additionally, there is a layer specific learnable weight matrix of dimension $ d_{in} \times d_{out} $ where $ d_{out} $ represents the output feature dimension. This strategy makes the number of trainable parameters ( $2\times d_{in} \times d_{in} + d_{in}\times d_{out}$ per layer) which is independent of the number of nodes in a GCN layer and hence, independent of the number of clips used for a video (Section B in the supplementary material). While fully connected graph convolutions will increase the computation with longer clip sequences, we can adopt sparse video sampling [2] or techniques like [3] to tradeoff computation.
>
> (c) **Effect of Background Extraction and Mixing Method**: We adopt a simple, intuitive, and computationally efficient Temporal Median Filtering (TMF) to extract the background which has proven to be successful and commonly adopted in several recent deep learning pipelines [4, 5] (see Section D in supplementary material for more details with qualitative examples). Moreover, as suggested by the reviewer, we experiment with a different background extraction strategy [6] that uses Gaussian Mixture Models (GMM) to extract the background and observe that the very simple and fast strategy based on TMF outperforms GMM by 2.3% on average on UCF-HMDB (UCF→HMDB: 85.3% vs 86.7%, HMDB→UCF:90.7% vs 93.9%, Avg: 88.0% vs 90.3%). While it would be interesting to study the effect of various learnable background segmentation strategies such as [7, 8], we would like to highlight that our CoMix framework is agnostic to the method used for background extraction and can be incorporated with any other background extraction techniques for videos.
>
> On the other hand, we also explore the effectiveness of the background mixing method by comparing with different MixUp strategies such as Gaussian Noise, Video MixUp and Video CutMix (Table 5 in the main paper).  The proposed way of generating synthetic videos by mixing the background of a video from one domain to a video from another domain, outperforms all three alternatives on UCF-HMDB (by a margin of about 3% on average).
>
> (d) **Effect of Target Pseudo-label Confidence Threshold**: Following reviewer’s suggestion, we study the the sensitivity of the final performance with respect to the pseudo label threshold on UCF-HMDB dataset and notice that CoMix performance is quite stable with respect to this parameter (best performance at threshold set to 0.7). The slight decrease in performance with PL = 0.9 is understandable since very few target videos are getting selected as additional positives for the supervised contrastive loss.
>
> \\begin{array} {|lccc|}
> \\hline \\text{PL Threshold} & \\text{U &#8594; H} & \\text{H &#8594; U} & \\text{Average} \\\\
> \\hline
> \\text{0.5}  & \\text{85.6} & \\text{93.5} & \\text{89.5} \\\\
> \\text{0.6}  & \\text{85.6} & \\text{93.5} & \\text{89.5} \\\\
> \\text{0.7}  & \\textbf{86.7} & \\textbf{93.9} & \\textbf{90.3} \\\\
> \\text{0.8}  & \\text{86.4} & \\text{92.5} & \\text{89.4} \\\\
> \\text{0.9}  & \\text{85.6} & \\text{90.9} & \\text{88.2} \\\\
> \\hline
> \\end{array}
>
> (e) **Alternatives to Graph Convolutional Network (GCN)**: Thank you for the suggestion. We replace GCN using MLP/LSTM and notice that both alternatives are inferior to GCN on UCF-HMDB dataset (as seen from the table below), which demonstrates the effectiveness of GCN in our contrastive learning framework for capturing the temporal dependencies, essential for video domain adaptation.
>
> \\begin{array} {|cccc|}
> \\hline \\textbf{Graph Alternatives} & \\textbf{U &#8594; H} & \\textbf{H &#8594; U} & \\textbf{Average} \\\\
> \\hline
> \\text{MLP}  & \\text{84.0} & \\text{92.2} & \\text{88.1} \\\\
> \\text{LSTM}  & \\text{84.2} & \\text{84.4} & \\text{84.3} \\\\
> \\text{GCN (Ours)}  & \\textbf{86.7} & \\textbf{93.9} & \\textbf{90.3} \\\\
> \\hline
> \\end{array}
>
> (f) **Missing References**: Thanks for pointing out the missing references on recent domain adaptation and mixup methods. We will include them in the final version of the draft.
>
> (g) **Minor Comments**: (a) We will correct the citation number for SAVA from [10] to [12] in the lines L276 and L308. (b) We will remove the redundancy in the introduction and proposed method (including figures) to make it clear in the final version. (c) The main motivation for Fig. 3 is to capture the overall CoMix framework. We appreciate the idea of including a more detailed diagram in the supplementary. We will add a detailed diagram including all the components in the supplementary.
>
> References :
> - [1] Multi-modal domain adaptation for fine-grained action recognition, CVPR 2020.
> - [2] Deep Analysis of CNN-based Spatio-temporal Representations for Action Recognition, CVPR 2021.
> - [3] Simplifying Graph Convolutional Networks, ICML 2019.
> - [4] BSUV-Net: A Fully-Convolutional Neural Network for Background Subtraction of Unseen Videos, WACV 2020.
> - [5] MotionRec: A Unified Deep Framework for Moving Object Recognition, WACV 2020.
> - [6] Improved adaptive Gaussian mixture model for background subtraction, ICPR 2004.
> - [7] Background Subtraction on Depth Videos with Convolutional Neural Networks, IJCNN 2018.
> - [8] Multi-frame Recurrent Adversarial Network for Moving Object Segmentation, WACV 2021.

---

> > ### Comment · Reviewer_bY6e · 2021-08-23
> > **Thank you for response.**
> >
> > I thank the authors for their efforts. The authors addressed my concerns, so I raised my score.

---

> > > ### Author Response · Authors · 2021-08-24
> > > **Thanks**
> > >
> > > Thanks for all the valuable feedback. We’re glad that our response addressed all your concerns.

---

### Official Review · Reviewer_UMhT · 2021-07-16

**Rating:** 6
**Confidence:** 4

**Summary:**

This work proposes a video-based contrastive learning framework by maximizing the similarity between encoded representations of the same video at two different speeds as well as minimizing the similarity between different videos played at different speeds. Results on UCF-HMDB, Jester, and Epic-Kitchens validate its effectiveness.

**Limitations And Societal Impact:**

This work has no potential negative societal impact.

**Main Review:**

[Merits]

1. No adversarial loss is needed, which helps to stabilize the training.

2. Formulating background mixing into the CL framework is novel.

3. Results on UCF-HMDB, Jester, and Epic-Kitchens as shown in Table 1-5 are good.

4. Code is provided.


[Improvement Suggestions]

1. How the background information is extracted?  Why In Line 65 "Importantly, since mixing background doesn’t change the temporal dynamics". Why? e.g. when the synthetic videos are generated via background mixing, what if it becomes false positive when the background information is added to the original frame?

2. This is an application of contrastive learning (Formula 3) with mixup (Formula 2) and pseudo labeling (Formula 4) in the scope of UDA for action recognition. Except for the second point (BG part in Formula 3) mentioned above, all components have been proposed in the previous work. There is no specific new method in terms of hard negative selection for the background. Could the author[s] justify the novelty of the proposed method precisely?

3. How long does it take for the SSL convergence? If it is stable across multiple seeds? Why only a single number is reported for all experiments? Table numbers look good, but the current form seems to lack ablation illustrations.

4. Could author[s] add discussions or results with the baseline mixup UDA method https://arxiv.org/pdf/2001.00677.pdf?

**Time Spent Reviewing:**

6 hours

---

> ### Author Response · Authors · 2021-08-10
> **Response to Reviewer UMhT**
>
> We thank the reviewer for the thoughtful reviews and great suggestions. Below are our responses to the concerns and will incorporate all the feedback in the final version.
>
> (a) **Background Information and Mixing**: We adopt a simple and fast Temporal Median Filtering (TMF) method for extracting background information which has proven to be successful and commonly adopted in several recent deep learning pipelines [1, 2]. Specifically, for videos, a pixel-wise temporal median filter is applied on the sequence of frames to obtain the corresponding background. The method is designed with the principle that for a given pixel location, in a sequence of frames, the most frequently repeated intensity along the temporal direction is most likely to be the background value for that pixel (Section D in supplementary material for more details including qualitative examples). Note that while we adopt this simple strategy for extracting backgrounds, our CoMix framework is agnostic to the method used for background extraction.
>
> The background mixing is actually performed by adding an extracted static background frame to the video (all of its frames). The addition of a static background frame to all frames of a video does not hinder the temporal action dynamics (motion patterns) possessed by the video. We validate this hypothesis by obtaining the optical flow for a given video before and after performing background mixing, and observe no significant change in them. Because of our inability to include images in the rebuttal, we could not provide them here. We will add these qualitative experiments to the supplementary of the final submission.
>
> (b) **Novelty of the Proposed Method**: Our key novelty is in introducing a new contrastive learning framework for learning domain invariant features, without requiring any additional adversarial learning, as most prior works do in video domain adaptation (which is often unwieldy to train as rightly pointed by the reviewer). This is achieved by using (a) speed invariance as temporal contrastive factor, (b) background mixing for exploiting action semantics shared across both domains and (c) target pseudo-labels to enhance discriminability of the latent space. In particular, we propose a simple yet effective extension to temporal contrastive loss (a), by using additional positives per anchor ((b) and (c)) in a supervised contrastive learning framework, which to best of our knowledge, has not been attempted to tackle the problem of video domain adaptation. Furthermore, while being simple, our approach achieves strong results and establishes new SOTA for unsupervised video domain adaptation, improving previous best results on three standard datasets.
>
> The Background mixing strategy and target pseudo labelling adopted in CoMix contribute to additional positives per anchor for the contrastive learning (Fig. 2), this observation is in accordance with a very recent work [5] that focuses on adding additional positive per anchor (using k-Nearest Neighbor) instead of hard negative mining. Employing hard negative mining has proven to be successful in a contrastive learning framework and using it on top of our CoMix framework should contribute to further performance improvement and would be an interesting aspect to explore in future.
>
> (c) **Convergence and Multiple Seeds**: Thanks for raising these. The convergence of the proposed approach varied with dataset and task complexity ranging from 3000 iterations for HMDB->UCF to 7000 iterations for UCF->HMDB and EpicKitchens. We observe that the convergence is fairly stable across different seeds and the performances reported in the paper are in fact the average over three runs with different random seeds (L296-297). In the submission, we did not report the standard deviations to avoid clutter. In absence of reported standard deviations from the other competing methods, a head-to-head comparison was not also imminent. The standard deviations for UCF-HMDB, Jester and EpicKitchens are 0.3, 0.1 and 0.2 respectively.
>
> (d) **Comparison with Baseline MixUp UDA Method [3]**: Thanks for pointing us to the reference on baseline MixUp method for improving unsupervised domain adaptation in images. We extend [3] to video domain adaptation on UCF-HMDB dataset and observe that CoMix outperforms this baseline by a margin of 8.9% on average on UCF-HMDB (UCF→HMDB: 77.2% vs 86.7%, HMDB→UCF:85.6% vs 93.9%, Avg: 81.4% vs 90.3%). We will add this experiment in the final version.
>
> **References**:
> - [1] BSUV-Net: A Fully-Convolutional Neural Network forBackground Subtraction of Unseen Videos, WACV 2020.
> - [2] MotionRec: A Unified Deep Framework for Moving Object Recognition, WACV 2020.
> - [3] Improve unsupervised domain adaptation with mixup training, arXiv:2001.00677.
> - [4] Improved adaptive Gaussian mixture model for background subtraction, ICPR 2004
> - [5] With a Little Help from My Friends:Nearest-Neighbor Contrastive Learning of Visual Representations, arXiv:2104.14548.

---

> > ### Comment · Reviewer_UMhT · 2021-08-21
> > **thanks for the response**
> >
> > Thank you for the response. The response address most of my questions. Response (a) and (b) shows that the proposed method is not a ad-hoc combination. Please consider add it into the text. I will raise my score.

---

> > > ### Author Response · Authors · 2021-08-22
> > > **Thanks for upgrading the score**
> > >
> > > We thank the reviewer for acknowledging that our proposed method is not an ad-hoc combination. We will add all the feedback/discussion in the final version.

---

### Official Review · Reviewer_LMAg · 2021-07-21

**Rating:** 6
**Confidence:** 4

**Summary:**

This paper aggregates (a) contrastive learning, (b) background mixing and (c) target pseudo-labels on video domain adaptation and achieves excellent performance (outperforming SOTA). The paper is written well and easy to follow.

**Limitations And Societal Impact:**

Please see limitations in the main review

Minor comments:
Figure 1 is too small to see how background is mixed

Figure 2 and 3 are not easy to follow.

Figure 5: please add more explanation on "become progressively discriminative".



**Main Review:**

The paper has solid results and ablations on several key datasets in video domain adaptation. The paper reads well.

However, the novelty is limited. Using contrastive learning to leverage labeled and unlabeled data (UDA is a special case) is not new. Sampling at different speeds and background mixing can be considered as data augmentation (SimCLR already showed that data augmentation is crucial for contrastive learning). In addition, using pseudo-labels in the target domain is not new either.

The idea of background mix is interesting, but it is not clear WHY it is more effective than Mixup or CutMix? Is it more effective in video domain adaptation? or it is a general approach for video recognition tasks.

Good performance is crucial for a paper, but I would like to see more insightful understanding in a NeurIPS paper. e.g. why sampling at different speeds is optimal for video domain adaptation, or it is just one of many working sampling strategies? is it dataset specific? why background mix is crucial for video domain adaptation? does background encode shortcut in the source domain?

**Time Spent Reviewing:**

4 hours

---

> ### Author Response · Authors · 2021-08-10
> **Response to Reviewer LMAg**
>
> Thanks for the thoughtful reviews and constructive suggestions.
>
> (a) **Technical Novelty**: Our key novelty is in introducing a new contrastive learning framework for learning domain invariant features, without requiring any additional adversarial learning, as most prior works do in video domain adaptation (which is often unwieldy to train). This is achieved by mainly using (a) speed invariance as temporal contrastive factor, (b) background mixing for exploiting action semantics shared across both domains and (c) target pseudo-labels to enhance discriminability of the latent space. In particular, we propose a simple yet effective extension to temporal contrastive loss (a), by using _additional positives per anchor_ ((b) and (c)) in a supervised contrastive learning framework (Fig. 2 in the main paper for more details), which to best of our knowledge, has not been attempted to tackle the problem of video domain adaptation. Since our work draws a new connection between unsupervised video domain adaptation and supervised contrastive learning, we believe our method is technically novel. Furthermore, while being simple, our approach achieves strong results and establishes new SOTA for unsupervised video domain adaptation, improving previous best results on three standard datasets.
>
> (b) **Background Mixing**: CoMix extracts one background per video from one domain and then mixes it with a video from another domain.  This enables us in creating new synthetic videos that have the same action semantics and possess the background of the other domain. In other words, the main operation in our background mixing is to generate a synthetic video with background from the other domain while retaining the temporal action semantics intact. Video MixUp and Video CutMix on the other hand, often destroy the motion pattern of the action in the original video due to straightforward addition of two actions from two different domains. Video MixUp and Video CutMix lead to an average performance of 88.4% and 88.3% respectively on UCF-HMDB, compared to 90.3% achieved by using background mixing (Table 5 in the main paper). We also test it on the challenging Jester dataset where our background mixing once again outperforms both Video MixUp and Video CutMix by more than 2% in accuracy (Video MixUp: 62.2%, Video CutMix: 58.6%, Background Mixing: 64.7%). This clearly suggests the importance of keeping action semantics consistency while generating synthetic videos.
>
> Our background mixing is especially effective in video domain adaptation as it enforces the model to be robust to domain changes (i.e., difference in background as shown in Figure 1) while leaving the action semantics intact. Our idea on background mixing can also be adopted as a data augmentation strategy for improved generalization in standard video action recognition: we leave this as an interesting future work. We will add these discussions in the final version.
>
> (c) **Sampling at Different Speeds**: Several data augmentation techniques have recently been used in self-supervised learning (e.g., SimCLR and its variants). However, to the best of our knowledge, we are the first to exploit implicit ‘time’ supervision (i.e., changing video speed does not change an action) in a contrastive learning framework for unsupervised video domain adaptation. Speed is an almost invariant feature for recognizing actions in many cases irrespective of the dataset (except a few examples such as walking vs. running or slow dance vs. fast dance, as noted by R-CMgF). While we agree that sampling videos can be regarded as a form of temporal data augmentation, our work provides new insights on how to exploit that rich supervisory information instead of the dominant adversarial learning for feature alignment in video domain adaptation.
>
> (d) **Figures**: Thank you for providing these suggestions. We will address the comments in Fig. 1, 2 and 3 to make them more clear in the final version. For Figure 5, the tSNE visualizations of the feature space of source [Blue] and target domain [Red] are shown. The clusters formed by using Source only models [very left] are not very discriminative and quite overlapping especially for the target domain. As opposed to that, the representation capability of the learned features in terms of distinct and tight-knit clusters is seen to get improved as we add the different components [TCL, BGM and TPL] one by one from left to right suggesting progressive discriminability.

---

> > ### Comment · Reviewer_LMAg · 2021-08-20
> > **Thanks for the response**
> >
> > Thanks for the answers, which address some of concerns.
> >
> > **Technical Novelty**: I am confused here, Is the novelty at the framework or each component. If the framework is the key innovation, I would like to see the evidence that a+b+c is necessary and their connections are mutually beneficial in terms of insights rather than just results. If any component is the key innovation, I would like to see the in-depth analysis for that component. This paper is currently more like an aggregation of three things together which achieves good results, but neither a framework nor a brand new algorithm.
> >
> > Let me use contrastive learning as an example, if this is a specific contrastive learning framework for video domain adaptation, what is the difference between this and contrastive learning for (a) image recognition, (b) image domain adaptation, and (c) video recognition, and why? If there exists a key difference, that may be an insightful finding.  It is hard to consider applying algorithm a (contrastive learning) to replace algorithm b (adversarial learning) in an application (domain adaptation) novel.
> >
> > **"Our background mixing is especially effective in video domain adaptation"** could you guild me where I can find the experiments that show background mixing is not effective in other video recognition tasks. Also, have you done any statistic analysis to show that background is the major noise for video domain adaptation?
> >
> > **the first to exploit implicit ‘time’ supervision:** if the speed augmentation is the key finding of this paper, reader would like to know if speed augmentation is crucial for all kinds of videos or just some types of videos. If it only works for some types of videos, what are those types (please show evidence)?

---

> > > ### Author Response · Authors · 2021-08-21
> > > **Response to Reviewer LMAg: Part 1**
> > >
> > > Thanks for the feedback! Please find below are our responses and let us know if you have any further questions/concerns. We would be more than happy to address them.
> > >
> > > (a) **Technical Novelty:** Our key technical novelty lies **both in individual components/losses and the framework.** Below we provide more details on the insights, innovation and empirical evidence on each component.
> > >
> > >
> > > - (i) **Temporal Contrastive Learning (TCL)**: **Insights:** Existing methods for unsupervised domain adaptation in videos [8, 12, 28, 43, 51, 54] and a vast majority of images [5, 21, 41, 56, 74] rely on adversarial learning (a minimax optimization problem) for feature alignment, which is widely known, both in theory and practice, to be very difficult to solve [a, b]. Unless carefully balanced, the push and pull in opposite directions can often cause wild fluctuations in the discrepancy loss and lead to sudden divergence resulting in very fragile convergence [71]. Motivated by this observation, one of the main goals of our work is to achieve alignment between the source and target domains by training a model on the same self-supervised contrastive task in both domains simultaneously. More specifically, we utilize temporal contrastive learning to get improved video representation to bridge the domain gap. This is done by maximizing the similarity between encoded representations of an unlabeled video at two different speeds as well as minimizing the similarity between different videos played at different speeds (Eq. 1). **Innovation:** Our key innovation in temporal contrastive learning is to employ implicit time supervision (i.e., changing video speed does not change an action irrespective of the domain) to achieve domain invariance and more importantly is to draw a connection between unsupervised domain adaptation and contrastive learning. **Empirical Evidence:** Our insightful innovation on temporal contrastive learning for easy domain invariance is suitably backed up with extensive experiments. Our vanilla temporal contrastive learning (without additional help from background mixing and target pseudo-labels) significantly outperforms both DANN and ADDA on both UCF-HMDB (2%: Table 4 in the main paper) and Jester datasets (5%: Table 4 in the supplementary material), showing its effectiveness over adversarial learning in aligning features.
> > >
> > >
> > > - (ii) **Background Mixing (BGM)**: **Insights:** Vanilla temporal contrastive loss treats each domain individually (Eq. 1). However, if temporal contrastive learning is equipped with the capability to focus more on the actual action semantics in the video and is able to ignore domain specific background details, it will be more useful for aligning the actions across domains. Thus our thought was to mix backgrounds of videos from different domains so that the model gets a chance to see the same action in a variety of backgrounds and thus learn to be indifferent to cross-domain background shift, as seen in Figure 1. However, one can not simply add vanilla contrastive learning with background mixing as mixing background doesn’t change the motion pattern of a video that actually defines an action (a straightforward combination will lead to incorrect positives and negatives per anchor). Thus, we modify the vanilla TCL loss (Eq. 1) to a supervised contrastive loss (Eq. 3) by considering the mixed video to be of the same class as the original video  (Figure 2). This encourages the model to generalize to new samples that may not be covered by the vanilla temporal contrastive learning in hand. In other words, mixed background video of an input sample act as small semantic perturbations that are not imaginary, i.e., they are representative of the action semantics shared across source and target domains. **Innovation:** Our main innovations in background mixing are: (a) incorporating dataset-specific knowledge into the design of contrastive learning, specifically for the application of video domain adaptation, and (b) formulating the problem as supervised contrastive learning (that allows additional positives per anchor) instead of self-supervised contrastive learning. Our formulation on background mixing into the contrastive learning framework is novel, as noted  by both Reviewer UMhT and Reviewer bY6e. **Empirical Evidence:** Three different sets of experiments clearly show the effectiveness of our new background mixing strategy on multiple datasets. First, background mixing improves the performance over vanilla contrastive learning (+2.9% in UCF-HMDB and +5.6% in Jester) and also leads to better features (as seen from the t-SNE visualizations in Figure 5). Second, background mixing outperforms three different alternatives (Gaussian Noise, Video MixUp and Video CutMix) on both UCF-HMDB and Jester, demonstrating its advantages over conventional MixUp strategies. Finally, the effect of pseudo-labels (i.e., additional positives per anchor) on background mixed videos in UCF-HMDB (Table 6) provides evidence on the effectiveness of our supervised contrastive learning framework in leveraging shared action semantics.
> > >
> > >
> > > - (iii) **Target Pseudo-Labels (TPL)**: **Insights:** While temporal contrastive loss with background mixing helps in aligning the learned representations across the two domains, we cannot fully rely on source categories to learn features discriminative for target domain. Thus, we generate pseudo-labels for the target samples in every batch and then harness the label information using a supervised contrastive term guided by these target pseudo-labels, that pushes the examples from the same class closer and the examples from different classes further apart (Eq. 4). **Innovation:** Our main innovation in TPL is in the formulation of supervised temporal contrastive loss by allowing more samples per anchor to be positive, so that videos of the same pseudo-label can be attracted to each other in the embedding space (Eq. 5). While pseudo-labeling by iteratively updating the classifier is shown to be helpful for the UDA in the image domain, we are not aware of any work in the literature that exploits TPL in a supervised contrastive learning framework, as we do for video domain adaptation. **Empirical Evidence:** Experiments on both UCF-HMDB (+1.9%) and Jester datasets (+1.6%) show that discriminability (through TPL) of the latent space plays a crucial role in video domain adaptation in addition to the vanilla contrastive learning for aligning features.
> > >
> > >
> > > - (iv) **Framework**: Our framework mainly consists of two key components (BGM and TPL as different losses in Eq. 6). The connection of these components with vanilla temporal contrastive learning (TCL) is non-trivial (as shown in Eq. 3 and Eq. 5) and only incrementally beneficial if used standalone (Table 4). More importantly, both are mutually beneficial as they focus on addressing two important yet distinct issues while learning invariant features for unsupervised video domain adaptation. For example, TPL helps to correctly assign positives per anchor that are considered as negatives with BGM; in Figure 2, TCL w/ BGM treats the video sample with action walking in snow as a negative sample for an anchor, while TCL w/ TPL considers the same video sample as an additional positive for the same anchor.
> > > We respectfully disagree with the reviewer’s statement that our paper is more like an aggregation of the three things together because of the above compelling reasons. Our key insights, innovation and strong empirical evidence clearly show that our paper is sufficiently novel (also stated by the other reviewers) in addressing a challenging and significant research topic of unsupervised domain adaptation for video recognition.
> > >
> > >
> > > **(b) Differences from Existing Contrastive Learning Frameworks:** We did not find any prior work on contrastive learning for unsupervised domain adaptation in image recognition. The only relevant work we are aware of that uses a notion of consistency for images is in domain-adaptive semantic segmentation [c], which is made publicly available in arXiv just around the NeurIPS submission deadline (17 May 2021, CVPR 2021). Our work is significantly different from [c] as we adopt instance-discrimination through contrastive loss on both source and target domains simultaneously, while [a] utilizes pixel-wise consistency between the model’s predictions only on target images. As for the existing works on contrastive learning for image/video recognition [9, 20], our work is different from them in two main aspects: (i) formulation: our framework presents a new perspective of supervised contrastive learning that utilizes additional synthetic positives per anchor (via BGM and TPL), as opposed to [9, 20] that uses only a single positive for self-supervised representation learning; (ii) scope of application: while [9, 20] focuses on conventional image/video recognition, our approach focuses on unsupervised video domain adaptation, motivated by the fact that one can achieve domain invariance through implicit time supervision across both domains simultaneously, alleviating the need for complex adversarial learning.
> > >
> > > We request the reviewer to consider the fact that our contrastive learning framework is specifically designed for video domain adaptation with novel components like BGM/TPL, which to the best of our knowledge, has not been studied in existing literature. We would be very grateful if the reviewer can provide any prior reference that adopts individual components/framework that are similar to ours in the literature.
> > >
> > > **(Please see next for the remaining response)**

---

> > > > ### Author Response · Authors · 2021-08-21
> > > > **Response to Reviewer LMAg: Part 2**
> > > >
> > > > **(c) Our background mixing is especially effective in video domain adaptation:** The background mixing is *originally proposed by us* for alleviating the cross-domain background shift in video domain adaptation. We are not aware of any prior work that proposes any such similar mixing strategy for either domain adaptation or any video understanding tasks. It is especially effective in video domain adaptation as it enforces the model to be robust to domain changes (i.e., difference in background) while leaving the action semantics intact. However, our idea on background mixing can also be adopted as a data augmentation strategy for improved generalization in standard video action recognition: we leave this as an interesting future work and we believe our work will invigorate stimulating research in this direction.
> > > >
> > > > Yes, we have done qualitative analysis on the UCF-HMDB dataset and observed that videos from two different domains with the same action differ mostly in background. This is a known problem in video domain adaptation, as also stated by Reviewer CMgF. We provide one such example in Figure 1, where videos in the top row are from the source and target domain respectively, but both capture the same action walking. Because of our inability to include images in the rebuttal, we could not provide them here. We will add additional qualitative examples to demonstrate the cross-domain background shifting to the supplementary of the final submission.
> > > >
> > > >
> > > > **(d) First to exploit implicit ‘time’ supervision:** We would like to clarify that we are not claiming speed augmentation as the key finding of this paper. Instead, we are the first to *exploit domain invariance through implicit time supervision* (i.e., changing video speed does not change an action irrespective of the domain) to achieve alignment between the source and target domains in domain-adaptive action recognition. Prior works show that speed augmentation is very effective in supervised [19], semi-supervised [68, 92] and self-supervised [1, 29, 77, 86] learning (L126-129), irrespective of the type of videos and datasets. Speed is an invariant feature for recognizing actions in many cases except a few examples such as walking vs. running or slow dance vs. fast dance, as also noted by Reviewer CMgF.
> > > >
> > > > - [a] How well can generative adversarial networks learn densities: A nonparametric view, arXiv preprint arXiv:1712.08244, 2017.
> > > > - [b] Minmax optimization: Stable limit points of gradient descent ascent are locally optimal. arXiv preprint arXiv:1902.00618, 2019.
> > > > - [c] PixMatch: Unsupervised Domain Adaptation via Pixelwise Consistency Training, CVPR 2021. Other references are from the main paper.

---

> > > > > ### Comment · Reviewer_LMAg · 2021-08-24
> > > > > **Thanks for the long response**
> > > > >
> > > > > Thanks authors for writing such a long response, I really appreciate this. However, I am more confused about the key contribution. **If only an innovation is claimed in a sentence, what is it?** This kind of focus is needed for a good NeurIPS paper.
> > > > >
> > > > > Also, I would suggest rephrasing "contrastive learning framework". Usually, a framework is general for multiple scenarios/tasks, like SimCLR, MoCo, BYOL. This paper is more like **applying contrastive learning** for video domain adaptation where the data augmentation is modified accordingly (background mixing for mitigating domain gap, and speed augmentation for video).

---

> > > > > > ### Author Response · Authors · 2021-08-24
> > > > > > **Followup Response**
> > > > > >
> > > > > > Thanks for the feedback! We are glad at your appreciation of our effort. Please find below our responses.
> > > > > >
> > > > > > - **Innovation in One Sentence:** Our key innovation is in introducing **a new temporal contrastive learning approach for unsupervised video domain adaptation**, which is achieved by *jointly* leveraging (a) video speed for domain invariance, (b) background mixing for exploiting action semantics and (c) target pseudo-labels for enhancing discriminability of the features.
> > > > > >
> > > > > > - **Application of Contrastive Learning:** Our work is not a straightforward application of contrastive learning for video domain adaptation.This is because of two main reasons. First, we introduce two novel components (BGM and TPL), specifically for addressing two important issues for unsupervised video domain adaptation (as discussed in our previous detailed response). Second, the two newly introduced components contribute to additional *synthetic positives per anchor*, which requires non-trivial modifications over the vanilla contrastive loss that uses only a single positive per anchor (Eq. 3 and Eq. 5). Note that a direct application of contrastive learning (i.e., TCL) leads to much worse performance on both UCF-HMDB (85.8% vs 90.3%) and Jester (57.5% vs 64.7%), clearly demonstrating the effectiveness of our formulation with the above significant changes for video domain adaptation.
> > > > > >
> > > > > > - **Notion of Framework:** Thanks for the suggestion. We will rephrase it in the final version.
> > > > > >
> > > > > > We hope that we are able to adequately address all of your confusions. If you have any further questions, please let us know. We would be more than happy to address them.

---

> > > > > > > ### Comment · Reviewer_LMAg · 2021-08-24
> > > > > > > **This is clearer, Thanks**
> > > > > > >
> > > > > > > Thanks for the quick response. This looks clearer.

---

> > > > > > > > ### Author Response · Authors · 2021-08-24
> > > > > > > > **Thanks**
> > > > > > > >
> > > > > > > > We are glad that our response could clarify all the interesting queries. Thanks a lot for the constructive discussion which, we are sure, made our paper stronger.

---

### Author Response · Authors · 2021-08-10
**Summary of Author Response**

We would like to thank all the reviewers for their constructive comments! We are encouraged that reviewers find that: (a) our work addresses a challenging and significant research topic of unsupervised domain adaptation for video recognition (R-bY6e), which has largely been unsolved (R-CMgF); (b) our formulation on background mixing into the contrastive learning framework is interesting (R-LMAg) and novel (R-UMhT, R-bY6e); (c) our method has a good motivation and sensible design choices (R-CMgF) with no requirement of adversarial loss, which helps to stabilize the training (R-UMhT); (d) our paper has solid results and ablations on several key datasets in video domain adaptation (R-LMAg, R-UMhT, R-bY6e).

We have addressed all the questions that the reviewers posed with additional experimental comparisons and clarifications. Below, we summarize the summary of the response and will incorporate all the feedback/changes in the final version.

- (a) Clarification and discussion on novelty, background mixing, sampling, and convergence, as suggested by R-LMAg and R-UMhT,
- (b) Comparison and discussion on MM-SADA, as suggested by R-bY6e and R-CMgF,
- (c) Comparison with baseline MixUp UDA method, as suggested by R-UMhT,
- (d) Additional ablation experiments on Jester dataset, as suggested by R-CMgF,
- (e) Baseline comparisons w/ GCN features, as suggested by R-CMgF,
- (f) Comparison with alternate temporal methods instead of GCN, as suggested by R-bY6e,
- (g) Effect of background extraction method and pseudo-label threshold, as suggested by R-bY6e.

---

### Author Response · Authors · 2021-08-19
**Request for feedback on the rebuttal**

Dear Reviewers,

We thank the reviewer's time for reviewing, and we really hope to have a further discussion with them to see if our response solves the concerns. We have addressed all the thoughtful questions and suggestions raised by the reviewers, and we hope that the work's impact and results are better highlighted with our responses. It would be great if the reviewers can kindly check our responses and provide feedback with further questions/concerns (if any). We would be more than happy to address them. Thank you!

Best wishes,

Authors

---

### Decision · Program_Chairs · 2021-09-27

**Decision:**

Accept (Poster)

**Comment:**

The initial reviews raised several concerns about novelty, missing experiments, and analyses that provide insights. The rebuttal addressed most of the concerns; some reviewers raised their ratings after the rebuttal. Overall, the paper shows how to combine existing ideas to tackle a challenging problem in a simple and effective manner. The reviews are generally positive, mainly due to the good empirical performance, but they are not overwhelmingly enthusiastic about the technical novelty. I carefully read all the reviews, rebuttal, and discussion, and also read the paper in detail. I tend to agree that the paper lacks novelty but must say the proposed approach is simple, effective, and performs well. I am recommending an acceptance, but wouldn't mind if it is rejected.